# AnnoSpat annotates cell types and quantifies cellular arrangements from spatial proteomics

Aanchal Mongia[1,2], Fatema Tuz Zohora[3,4], Noah G. Burget[1,2], Yeqiao Zhou[1,2], Diane C. Saunders [5], Yue J. Wang [6], Marcela Brissova[5], Alvin C. Powers[5,7,8], Klaus H. Kaestner [2,6,9], Golnaz Vahedi [2,6,9], Ali Naji[9,10], Gregory W. Schwartz [3,4,11] ✉ & Robert B. Faryabi [1,2] ✉

Cellular composition and anatomical organization influence normal and aberrant organ functions. Emerging spatial single-cell proteomic assays such as Image Mass Cytometry (IMC) and Co-Detection by Indexing (CODEX) have facilitated the study of cellular composition and organization by enabling high-throughput measurement of cells and their localization directly in intact tissues. However, annotation of cell types and quantification of their relative localization in tissues remain challenging. To address these unmet needs for atlas-scale datasets like Human Pancreas Analysis Program (HPAP), we develop AnnoSpat (Annotator and Spatial Pattern Finder) that uses neural network and point process algorithms to automatically identify cell types and quantify cell-cell proximity relationships. Our study of data from IMC and CODEX shows the higher performance of AnnoSpat in rapid and accurate annotation of cell types compared to alternative approaches. Moreover, the application of AnnoSpat to type 1 diabetic, non-diabetic autoantibody-positive, and non-diabetic organ donor cohorts recapitulates known islet pathobiology and shows differential dynamics of pancreatic polypeptide (PP) cell abundance and CD8[+] T cells infiltration in islets during type 1 diabetes progression.

Tissues consist of diverse cell types whose functions are influenced by communication and interaction with surrounding cells. In addition to cell intrinsic aberrations, dysfunction in the cellular microenvironment impacts organ function and contributes to pathology of complex diseases, such as type 1 diabetes (T1D). The emergence of spatially resolved single-cell proteomic assays such as Image Mass Cytometry (IMC) and Co-Detection by Indexing (CODEX) has allowed high-throughput measurement of cellular composition and localization within intact tissues and advanced understanding of intricate cell–cell interactions. However, the unique characteristics of spatial proteomic assays, coupled with their ability to measure millions of cells, have created a need for efficient and automated computational tools that

[1]Department of Pathology and Laboratory Medicine, University of Pennsylvania Perelman School of Medicine, Philadelphia, PA, USA. [2]Epigenetics Institute, University of Pennsylvania Perelman School of Medicine, Philadelphia, PA, USA. [3]Princess Margaret Cancer Centre, University Health Network, Toronto, ON, Canada. [4]Vector Institute, University of Toronto, Toronto, ON, Canada. [5]Department of Medicine, Vanderbilt University Medical Center, Nashville, TN, USA. [6]Department of Genetics, University of Pennsylvania Perelman School of Medicine, Philadelphia, PA, USA. [7]Department of Molecular Physiology and Biophysics, Vanderbilt University, Nashville, TN, USA. [8]VA Tennessee Valley Healthcare System, Nashville, TN, USA. [9]Institute for Diabetes, Obesity and Metabolism, University of Pennsylvania Perelman School of Medicine, Philadelphia, PA, USA. [10]Department of Surgery, University of Pennsylvania Perelman School of Medicine, Philadelphia, PA, USA. [11]Department of Medical Biophysics, University of Toronto, Toronto, ON, Canada. ✉e-mail: Gregory.Schwartz@uhn.ca; faryabi@pennmedicine.upenn.edu

enable identification of cell types and quantification of their spatial colocalization.

Despite the scarcity of algorithms for cell-type annotation from IMC and CODEX data, several approaches have been proposed to predict cell types from single-cell RNA sequencing (scRNA-seq) data[1]. Many of these methods, such as scmap and Garnett, employ clustering to group together transcriptionally similar cells and then map each cluster to reference cell types from a priori annotated datasets using representative cells from each group[2,3]. These methods rely on accurate clustering and reference data annotation, which was previously characterized based on manual assessment of differential expression of selected marker genes. Another category of scRNA-seq cell-type annotators utilizes supervised machine learning models, including support vector machines (SVM)[4], neural networks[5], and random forests[6,7]. A third category includes similarity-based methods like TooManyPeaks[8], which annotate cell types based on bulk measurement of purified reference cell populations. The training of supervised machine learning- and similarity-based methods necessitates large sets of purified or expert-annotated cell populations, which are lacking for IMC and CODEX in situ proteomic assays.

The unique characteristics of IMC and CODEX data further limit the applicability of existing cell-type annotation methods developed for scRNA-seq. While scRNA-seq experiments provide expression data for thousands of genes for cell-type prediction, IMC and CODEX measure the expression of tens of proteins. Moreover, IMC and CODEX readouts consist of continuous intensities that cannot be readily used with most scRNA-seq cell-type annotators like Garnett, which only accept scRNA-seq count data. To address these limitations, Astir was recently proposed as a dedicated method for cell-type annotation from IMC data[9]. This method employs deep recognition neural networks to infer cell types based on known marker proteins. Benchmarking studies of Astir suggest that supervised- and marker-based cell-type annotation methods tend to outperform other approaches[9].

Cell-type annotation is an initial step in the analysis of spatial proteomic IMC and CODEX data. To fully leverage the capabilities of in situ single-cell assays and explore the tissue microenvironment, methods are needed to quantitatively assess the spatial organization of cells within regions of interest. Existing approaches include measuring cell density over distances[7], employing Bayesian models for estimating cell types across locations[10], and utilizing Ripley statistics[11]. However, these algorithms can only assess the distribution of a single cell type within a region of interest (ROI), as seen in Ripley's $K$ function statistics, and cannot examine the proximity of multiple cell types.

In this work, we introduce AnnoSpat (Annotator and Spatial Pattern Finder): a comprehensive tool that addresses the unmet need for rapid, scalable, and automated annotation of cell types and quantification of their spatial relationships in atlas-scale IMC and CODEX datasets. AnnoSpat integrates semi-supervised and supervised learning methods to enable automated cell-type annotation from IMC and CODEX data in the absence of manually labeled cells for training. AnnoSpat is also equipped with new point process-based algorithms, which allow the quantification of spatial relationships among multiple cell types.

We evaluate the accuracy and efficiency of AnnoSpat by conducting benchmark tests to assess its capability to identify various cell types within pancreatic tissues. In addition to our benchmarking analyses with IMC and CODEX data, we assess the concordance between expert and AnnoSpat cell-type labeling in pancreata from T1D and non-diabetic donors. Pancreas is the site of T1D pathogenesis in which the host immune system mounts a response to insulin-secreting pancreatic beta cells. To further evaluate AnnoSpat's ability to recapitulate known changes in the pancreatic microenvironment during T1D progression, we analyze pancreata from (AAb$^+$) donors – individuals with autoantibodies toward pancreatic islet proteins in their blood but no clinical diagnosis of T1D. Together, our comprehensive analysis of 1,170,000 cells from 143 slides of 19 Human Pancreas Analysis Program (HPAP) donors reveals the effectiveness of AnnoSpat in reliably identifying cell types and quantifying their expected spatial organization in complex tissues. AnnoSpat and its individual components are available through https://github.com/faryabiLab/AnnoSpat.

## Results

### AnnoSpat identifies cell types and quantifies their relative localization

To predict the identity of individual cells and quantify their localization within tissues, we developed AnnoSpat for automated analysis of spatially aware single-cell proteomic data (Fig. 1). AnnoSpat provides an end-to-end solution for analysis of IMC and CODEX data (Fig. 1a) by implementing two distinct but complementary functionalities: Annotator (Fig. 1b) and Spatial Pattern Finder (Fig. 1c).

To address the unmet need for annotating individual IMC- or CODEX-measured cells, the Annotator module of AnnoSpat learns a cell-type predictor from the matrix of raw protein expression levels and a list of a priori cell-type marker proteins. To overcome the lack of manually annotated training data, AnnoSpat implements a two-step learning process (Fig. 1b). First, AnnoSpat deploys a constrained K-means semi-supervised clustering (SSC) algorithm to create training data from a subset of cells in the dataset. Using this automatically generated training data, AnnoSpat then trains a classifier that will be used to predict the identity of additional cells. The number of clusters is set to the number of expected cell types within the tissue of interest along with an optional Unknown group that could account for cell types omitted from the marker protein list (Maker Protein file). To enhance the accuracy of K-means clustering, AnnoSpat initializes each cluster with cells that were annotated with high confidence based on the distinct expression of marker proteins (see "Methods"). This crucial step provides semi-supervision to the clustering algorithm, guiding AnnoSpat in grouping a subset of cells with similar protein expression levels into cell-type-labeled training cells. Taking this automatically labeled data, AnnoSpat then trains an extreme learning machine (ELM) classifier. ELM is a feed-forward neural network with non-iterative single-step learning, which does not require tuning and backpropagation, and provides generalization performance and orders of magnitude faster learning compared to SVM and multi-layer perceptron[12] (see "Methods"). Together, the two-step learning algorithm equips AnnoSpat with an efficient and accurate cell annotation mechanism.

To facilitate the study of tissue microenvironment, we equipped AnnoSpat with the Spatial Pattern Finder module, which takes as input the Annotator-predicted cell types and their physical coordinates on the tissue ROI and quantifies cellular localization patterns (Fig. 1c). The Spatial Pattern Finder algorithm applies point process theory to summarize cell relationships across a range of distances, from local neighborhoods to remote cells. Briefly, AnnoSpat compares cell pairs based on their cell type to any randomly chosen cells at a given distance apart. This process returns a mark cross-correlation function, a measure of cell-type aggregation at different distances (see "Methods"). The application of the mark cross-correlation function across ROIs allows for systematic quantification and comparison of inter-cell-type proximity in different conditions (Fig. 1c). In addition to AnnoSpat software, we implemented Spatial Pattern Finder within the TooManyCells single-cell analysis suite[13]. This implementation includes the generation of interactive proximity plots that may be filtered by protein expression to fine-tune cell-type annotation. These interactive features also assist with the exploration of spatial cell relationships. AnnoSpat's Annotator and Spatial Pattern Finder functionalities together provide a solution for rapid and accurate annotation of millions of cells to study tissue microenvironment and cellular organization.

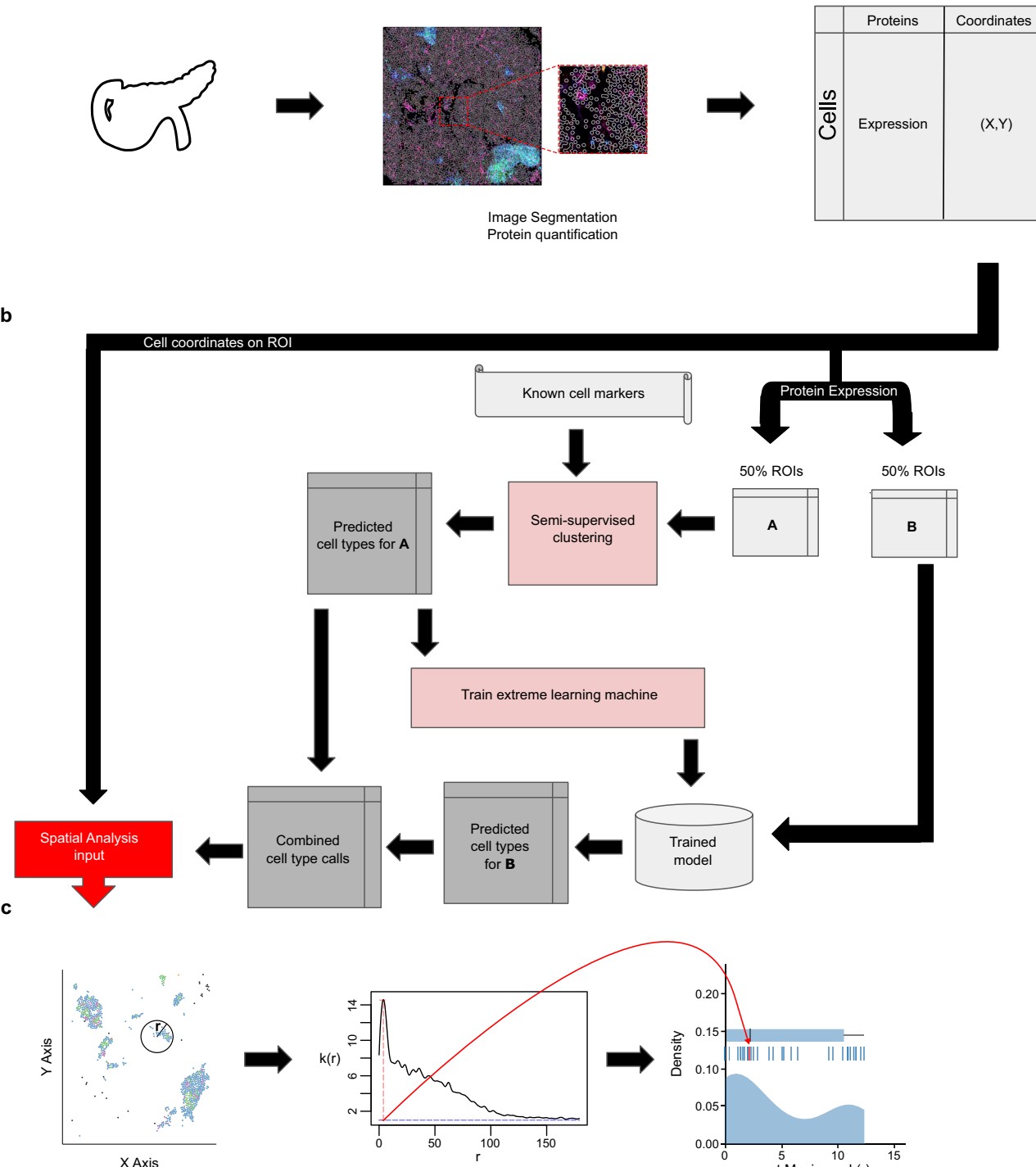

**Fig. 1 | Overview of IMC or CODEX data analysis with AnnoSpat (Annotator and Spatial Pattern Finder). a** From left to right: a tissue's region of interest (ROI) (e.g., from the pancreas) is measured using a spatial single-cell proteomics assay such as IMC or CODEX, reporting position and protein expression levels of individual cells in situ, which depend on cell segmentation and channel intensity quantification (see "Methods"). **b** To overcome lack of manually annotated training data, AnnoSpat's Annotator module learns a cell-type predictor by first processing protein expression data with a semi-supervised clustering algorithm, which creates a training dataset from a subset of cells in the overall dataset (e.g., 50% in matrix $\mathbb{A}$). Using this automatically generated training data, AnnoSpat then trains and applies an extreme learning machine classifier to label the remaining cells (e.g., 50% in matrix $\mathbb{B}$). **c** AnnoSpat's Spatial Pattern Finder component interprets cell locations as point processes to quantify relationships between cell types using distance-dependent ($r$) mark cross-correlation function ($k(r)$). Mark cross-correlation functions across ROIs are systematically summarized using different features of them such as the distance where the function is maximal.

## Comparison between AnnoSpat and supervised methods through cell-type identification in IMC-measured pancreatic tissues

To assess AnnoSpat's Annotator performance, we used IMC experiments measuring 33 proteins (Supplementary Data 1, 2), which can ideally distinguish up to 16 different cell types in pancreata from T1D and non-diabetic HPAP donors. To this end, we compared the ability of AnnoSpat, our SSC, SCINA, AUCell, and Astir to identify endocrine cell types. We considered these methods for comparative analysis since, similar to AnnoSpat, they automate cell-type annotation and do not need training data. Astir uses a probabilistic Bayesian framework and is specifically developed for cell-type annotation from proteomics data and so is the most comparable to AnnoSpat[9]. SSC is a variant of AnnoSpat with its classifier replaced by centroids from the SSC step in Fig. 1b (see "Methods"). SCINA[14] and AUCell[15] use expectation-maximization and gene expression ranking for cell-type annotation from scRNA-seq count data, respectively. We used default or suggested filters and parameters for all algorithms except AUCell, where size-factor normalization was disabled due to differences between the characteristics of discrete scRNA-seq count and continuous IMC data. AnnoSpat and SSC both used the Maker Protein file in Supplementary Data 3 as input.

In the absence of true cell identity labels, we first examined the extent of protein expression homogeneity in cell types predicted by these supervised cell-type labeling methods. To this end, we calculated the Silhouette Index (SI) on ten sets of 50,000 randomly selected cells (Fig. 2a–c, row 1; Supplementary Data 4–6). SI assesses how a cell's protein expression differs from other cells assigned the same label versus those assigned other labels. SI quantifies the extent of protein expression homogeneity in cells with a given label but not the correctness of their labels. AUCell-labeled alpha cells in samples from T1D donors, where alpha cells are abundant, had slightly higher mean SI than AnnoSpat. In contrast, AUCell had the lowest SI for beta-labeled cells in samples from T1D donors, where immunological destruction of beta cells results in low beta-cell abundance (Fig. 2a, row 1), suggesting additional challenges when detecting rare cells from IMC data. Similarly, most methods underperformed in detecting the epsilon cells as reported by our SI analysis, which is also a rare endocrine cell type in islets (Fig. 2a–c, row 1). SCINA- and AUCell-annotated pancreatic polypeptide (PP) cells were more homogenous than their respective labeled delta cells. AnnoSpat- and SSC-labeled delta cells in the control samples were more homogenous compared to other algorithms (Fig. 2b, row 1). SCINA, designed for scRNA-seq count data, underperformed in T1D, Control, and T1D plus control (Combined) cohorts (Fig. 2a–c, row 1), underscoring the need for cell-type calling algorithms specifically designed for spatial proteomics data that is fundamentally different from scRNA-seq count data. Importantly, Astir, which is developed for cell-type detection from IMC data, showed lower performance in identifying homogenous group of cells for many cell types in both control and T1D samples (Fig. 2a–c, row 1). In summary, our SI analysis revealed that cell-type annotation methods have cell-type and disease-state-dependent capability in identifying cells with homogenous protein expression profiles.

To complement SI analysis and further benchmark the correctness of cell-type annotation algorithms without true cell-type labels, we inspected protein expression profiles of endocrine-labeled cells in T1D, Control, and Combined cohorts (Fig. 2a–c, rows 2–6; Supplementary Fig. 1a–c, rows 1–5). Like any other antibody-based assay, IMC data quality highly depends on the sensitivity and specificity of antibodies used in the assay. Compared to other cell types, alpha, beta, PP, delta, and epsilon cells were particularly suitable for comparative benchmarking analysis due to the higher specificity of their antibodies in the HPAP IMC assay. We used a variant of term frequency-inverse document frequency (TF-IDF) normalization to reduce the effect of

non-specific antibodies such as anti-CD99 and anti-beta actin on data visualization (Fig. S4e–g; Supplementary Fig. 2a–c and "Methods").

Inspection of protein expression profiles of cells annotated as alpha, beta, PP, delta, and epsilon from IMC of T1D and non-diabetic donors (Fig. 2a–c, rows 2–6; Supplementary Fig. 1a–c, rows 1–5) showed the higher performance of AnnoSpat compared to SCINA and Astir. SCINA- and Astir- but not AnnoSpat-predicted beta cells from T1D samples, where beta cells are rare, showed high levels of immune cell-restricted proteins CD57 and HLA-ABC (Fig. 2a–c, rows 2, 5, and 6; Supplementary Fig. 1a–c, rows 1, 4, and 5). Comparing the result of cell-type prediction in T1D and Combined cohorts showed that additional samples improved the performance of AnnoSpat more so than Astir. Notably, Astir equally failed to detect epsilon cells in T1D, Control, and Combined cohorts (Fig. 2a–c, row 5; Supplementary Fig. 1a–c, row 4). CD11b, a marker of dendritic cells, was the highest expressed protein in the Astir-predicted delta cells (Fig. 2a–c, row 5; Supplementary Fig. 1a–c, row 4). Furthermore, Astir-predicted alpha cells expressed high levels of somatostatin, a canonical marker of delta cells (Fig. 2a–c, row 5; Supplementary Fig. 1a–c, row 4). Similar to Astir, SCINA failed to detect delta cells in samples from non-diabetic donors (Fig. 2a–c, row 6; Supplementary Fig. 1a–c, row 5). Moreover, SCINA-annotated PP cells were less homogeneous compared to the AnnoSpat-labeled cells (Fig. 2a–c, rows 2 and 6; Supplementary Fig. 1a–c, rows 1 and 5). Together, this data complemented our homogeneity benchmarking (Fig. 2a–c, row 1) and showed higher or comparable performance of AnnoSpat compared to other supervised cell-type annotation algorithms.

In addition to endocrine cells, AnnoSpat effectively detected other cell types that had high-quality antibodies in HPAP IMC assay and are commonly present in the pancreatic tissue (Supplementary Figs. 2c–h and 4e–g). For instance, AnnoSpat, SSC, and AUCell but not Astir, and SCINA clearly identified CD8+ T cells that had a specific antibody (Supplementary Figs. 2c–g and 4e–g). Conversely, the detection of helper and memory T cells was less accurate due to their less specific antibodies (Supplementary Figs. 2c–g and 4e–g). To further evaluate the impact of HPAP IMC panel's non-specific antibodies on the performance of cell-type annotation, we quantified protein expression heterogeneity with the SI metric and inspected protein expression profiles of cells annotated as CD8+ T and regulatory T cells. SI analysis and protein expression profiles concordantly showed that AnnoSpat and SSC outperform other algorithms in identifying CD8+ T cells (Supplementary Figs. 2c–g and 4h), which had a specific antibody in the HPAP IMC assay (Supplementary Fig. 4e–g). SI analysis also suggested that Astir and AnnoSpat equally outperformed other cell-type annotators in annotating regulatory T cells (Supplementary Fig. 4h). However, close examination of protein expression profiles revealed that Astir-labeled and AnnoSpat-labeled regulatory T cells had high levels of beta-actin and pS6 expressions, respectively (Supplementary Fig. 2c, f). Together, this analysis shows that the specificity of antibodies used in spatial proteomic assay impacts the ability of algorithms to accurately annotate cell types, as expected, and hence should be accounted for in the design of benchmarking experiments and during cell-type annotation.

## Comparison between AnnoSpat and unsupervised methods through cell-type identification in IMC-measured pancreatic tissues

Besides supervised cell-type predictors, we compared AnnoSpat with K-means, Seurat[16], PhenoGraph[17], and FlowSOM[18] clustering-based cell-type annotators due to their popularity in scRNA-seq and CyTOF data analysis (Fig. 1a–c, row 7; Supplementary Fig. 1a–c, rows 3, 4, and 6–9).

FlowSOM failed to effectively separate the cells in Combined, Control, and T1D cohorts leaving 1,151,979, 793,571, and 363,961 cells in a single cluster, respectively. For instance, FlowSOW labeled only

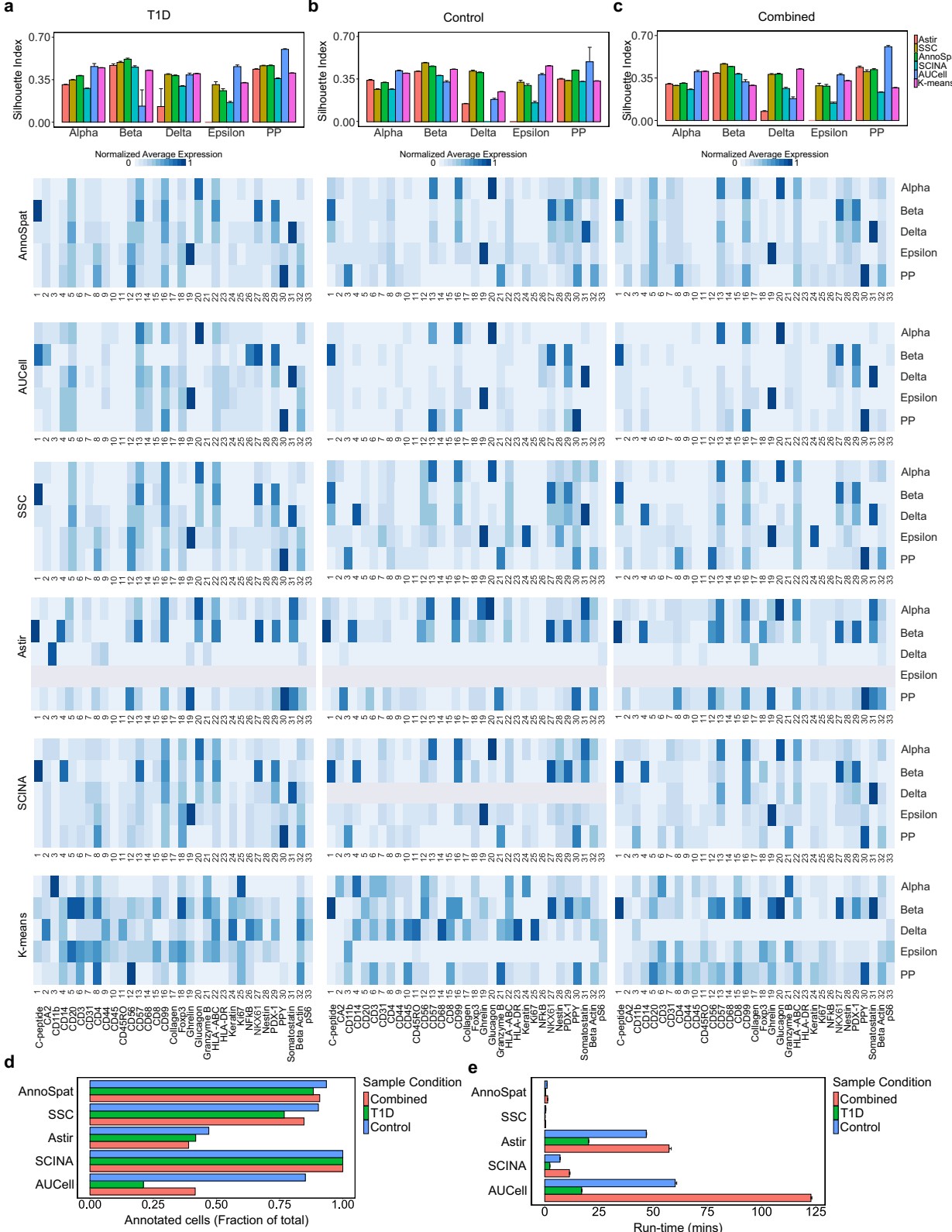

84, 259, and 2109 cells as alpha cells in Combined, Control, and T1D cohorts, respectively, which is not biologically plausible.

SI analysis showed that Seurat, one of the most popular scRNA-seq analysis tools, had the worst performance among the clustering-based cell-type annotators in identifying homogenous group of endocrine-labeled cells (Supplementary Fig. 3, row 1; Supplementary Data 4–6), underscoring differences between single-cell proteomics and transcriptomics data.

While both PhenoGraph and Seurat are based on Louvain clustering, they showed distinct SI values in clustering IMC data (Supplementary Fig. 3, row 1; Supplementary Data 4–6), potentially due to differences in normalization and/or Louvain algorithm

**Fig. 2 | Comparative analysis of AnnoSpat cell-type annotation from IMC data.**
**a** From top to bottom: bar plots with error bars showing average and standard deviation Silhouette Index (SI), heatmaps showing normalized average expression of all the 33 HPAP IMC-measured proteins for cells annotated as alpha, beta, delta, epsilon, and PP by AnnoSpat, AUCell, our semi-supervised clustering (SSC), Astir, SCINA, and K-means from T1D pancreas IMC data (*n* = 374,397 measured cells). **b** Similar to (**a**) from non-diabetic (control) pancreas IMC data (*n* = 795,604 measured cells). **c** Similar to (**a**) from combined T1D and control pancreas IMC data (*n* = 1,170,001 measured cells). *m* = 20 sets of *n* = 50,000 randomly selected cells

are used for evaluation using SI in each bar plot in the top panel of **a**–**c**. **d** Bar plots showing the fraction of *n* = 374,397, *n* = 795,604, and *n* = 1,170,001 IMC-measured cells form T1D, control, and combined T1D and control pancreata, respectively, annotated by AnnoSpat, SSC, Astir, SCINA, and AUCell. **e** Bar plots with error bars showing the mean and standard deviation of run-time for the listed algorithms to annotate cells as in (**d**). Each algorithm was run 15 times on a machine with Ubuntu 20.04, 1.05TB Memory, Intel Xeon Gold CPU 6230R @ 2.1GHz, 2 physical processors 52 cores, and 104 threads. Processed IMC data used in all the figures are provided as a Source Data file.

implementation. Notably, PhenoGraph, which is designed for CyTOF data, resulted in higher SI values compared to Seurat. Nonetheless, AnnoSpat's labeling resulted in mean SI values at least 0.1 higher than those of PhenoGraph in 5 out of 15 (cell type, cohort) comparisons between the two methods. PhenoGraph's mean SI values were only close to those from AnnoSpat in T1D and Control epsilon-labeled cells (Supplementary Fig. 3, row 1; Supplementary Data 4–6). In the absence of true cell-type labels, we again looked at the protein expression profiles of these cells to discern the accuracy of cell-type labeling (Supplementary

Fig. 3, row 3). While AnnoSpat-labeled epsilon cells clearly expressed ghrelin, the canonical marker of epsilon cells, CD11b (a canonical marker of myeloid cells) was the only distinctly expressed protein in the PhenoGraph-labeled epsilon cells (Fig. 2a, b; Supplementary Fig. 3a, b, row 1). However, the cells grouped and labeled as epsilon by PhenoGraph did not distinctly express ghrelin, the canonical marker of epsilon cells, hence, this group of PhenoGraph-labeled epsilon cells are indeed not biologically defined epsilon cells.

Similar to PhenoGraph, K-means analysis revealed that mean SI values were lower or close to those of AnnoSpat for all the endocrine-labeled cells in the Control cohort, except the K-means-labeled epsilon cells (Fig. 1; Supplementary Fig. 3, row 1, and Supplementary Data 4–6). Similar to PhenoGraph results, CD11b, not ghrelin, was the only distinctly expressed protein in the K-means-labeled epsilon cells (Supplementary Fig. 1, row 3), suggesting incorrect cell-type annotation by K-means despite higher SI. This analysis further supports the inability of K-means and Louvain-based clustering methods to correctly segregate rare populations in IMC data, an observation that was reported for scRNA-seq and scATAC-seq data[8,13]. AnnoSpat attempts to overcome this issue by implementing a procedure that initializes cluster centroids based on representatives of each cell type (see "Methods"). Examination of cell identities based on biological knowledge further showed that similar to AnnoSpat most supervised methods included in our analysis outperformed K-means-based cell-type prediction. For instance, somatostatin and PPY were highly expressed in AnnoSpat-, SSC-, SCINA-, and AUCell-labeled, but not in K-means-labeled delta and PP cells, respectively (Fig. 2a–c, row 7 versus other rows; Supplementary Fig. 1a–c, row 6 versus rows 1–5). Similarly, the unsupervised K-means clustering approach failed to accurately label alpha cells in T1D, Control, and Combined cohorts. Instead of high levels of glucagon, K-means clustering-labeled alpha cells showed high levels of CD11b, CD14, and granzyme B in T1D, Control, and Combined cohorts, respectively (Fig. 2a–c, row 7; Supplementary Fig. 1a–c, row 6). K-means, PhenoGraph, and Seurat also did not effectively label a subset of non-endocrine cells with high-quality antibodies in the HPAP IMC assay. For instance, in contrast to AnnoSpat, K-means-, PhenoGraph-, and Seurat-labeled CD8+ T cells were not true CD8+ T cells, because they did not distinctly express CD8 (Supplementary Figs. 2c and 4b–d). Together, our AnnoSpat comparative analysis with K-means, Seurat, PhenoGraph, and FlowSOM shows that while these unsupervised methods can produce comparable or more homogeneous groups of cells, they fail to segregate the cells based on their true lineage potentially due to stochastic initialization and/or IMC assay resolution, noise, and artifacts.

## Impact of initialization on AnnoSpat performance

To assess the generalizability of AnnoSpat-trained model, we evaluated the impact of AnnoSpat's initialization on cell-type annotation. To this end, we performed two experiments examining how the exclusion or inclusion of cell types in the initialization step affects AnnoSpat's performance. First, we removed NK and myeloid cells from the Marker Protein file (Supplementary Data 3) and predicted the identity of cells in the Combined cohort. Second, we added myeloid cells back to the AnnoSpat's Marker Protein file and repeated the cell-type annotation. The first analysis showed that the removal of these two cell types from AnnoSpat's initialization has a marginal impact on its performance in detecting cell types with specific antibodies (Supplementary Fig. 5a): cells annotated as alpha, beta, delta, PP, epsilon, acinar, and ductal still expressed high levels of respective marker proteins (glucagon, C-peptide, somatostatin, PPY, ghrelin, pS6, and CA2) in the absence of NK and myeloid cells from AnnoSpat's initialization. Sankey plots further corroborated this analysis and showed that removal of myeloid and NK from AnnoSpat's initialization had a marginal impact on predicted cell labels with specific antibodies and resulted in relabeling of almost all the myeloid and NK cells as Unknown (Supplementary Fig. 5c). The second analysis showed that inclusion of a cell type to AnnoSpat's initialization also had a limited impact on its performance (Supplementary Fig. 5b, d). We observed that the inclusion of myeloid markers to AnnoSpat's Marker Protein file relabeled some of the Unknown cells as myeloid, which had high levels of CD68 expression as is expected from a correct cell-type prediction.

To further evaluate the impact of AnnoSpat's initialization on performance, we examined how changes in cluster centroid initialization by $q_{high}$ impact protein expression profiles of annotated cell types (Supplementary Fig. 6a–j). In line with recommended default parameters (see "Methods"), the impact of $q_{high}$ on performance depended on cell-type abundance and antibody specificity. This analysis showed that a more relaxed $q_{high}$ (i.e., lower values) decreased the accuracy of cell-type prediction for both abundant and rare cell types, with a greater impact on rare cell populations (Supplementary Fig. 6c, d, h–j). For instance, using a more relaxed $q_{high}$ led to markedly lower levels of C-peptide and ghrelin in cells predicted as beta and epsilon in the Combined cohort, respectively. In contrast, a more stringent $q_{high}$ (i.e., higher values) had a marginal impact on performance, especially for abundant cell types, even with moderate antibody specificity, like acinar cells (Supplementary Fig. 6a–c, f–h). Together, this analysis clarifies how expected cell population abundance could guide AnnoSpat initialization by showing that too relaxed and stringent $q_{high}$ values could impact cell-type accuracy by potentially including non-representative cells and measurement artifacts in a training dataset, respectively.

## AnnoSpat can accurately predict cell types in new IMC samples

We further assessed the generalizability of AnnoSpat-trained model by predicting the identification of cells in completely independent experiments. To this end, we used the AnnoSpat-trained model to annotate cells in two new ROIs from the most recent HPAP donors that were not part of the original cohort listed in Supplementary Data 2. This analysis clearly demonstrated the ability of AnnoSpat to accurately predict cell types with specific antibodies in the HPAP IMC assay

(Supplementary Fig. 7a). Inspection of protein expression profiles of AnnoSpat-annotated alpha, beta, delta, PP, epsilon, and CD8⁺ T cells in these two new ROIs showed high levels of glucagon, C-peptide, somatostatin, PPY, ghrelin, and CD8 expressions, respectively.

## AnnoSpat rapidly annotates large fractions of cells in IMC samples

In addition to accuracy, we compared the completeness and run-time of cell-type annotation. By using ELM, AnnoSpat annotated more than 90% of 1.1 million cells (Fig. 2d; Supplementary Data 7) in <2 min, a run-time only 3 times longer than our SSC method and notably faster than all other supervised algorithms (Fig. 2e; Supplementary Data 8). Due to manual and laborious procedure, cell annotation with unsupervised methods is in the order of days and not minutes; hence, they were excluded from the execution time benchmarking. Although SSC and AnnoSpat mostly exhibited comparable performance, close examination of data highlighted the additional benefit of AnnoSpat (Fig. 2a–c, rows 2, 4; Supplementary Figs. 1a–c, rows 1, 3, and 2c, e). For instance, SSC- but not AnnoSpat-annotated delta cells expressed high levels of CD14, a protein expressed in macrophages and not delta cells (Fig. 2b, c, rows 2, 4; Supplementary Fig. 1b, c, rows 1, 3). Notably, Astir failed to label nearly 50% of the cells (Fig. 2d; Supplementary Data 7) while took 40 times longer (Fig. 2e; Supplementary Data 8). Due to its bi-modal distribution model, SCINA assigned a label to almost all the cells in a reasonable time (Fig. 2d, e; Supplementary Data 7, 8) at the expense of diminished accuracy (Fig. 2a–c, rows 2, 6; Supplementary Fig. 1a–c, rows 1, 5). Conversely, AUCell exhibited comparable performance to AnnoSpat (Fig. 2a–c, rows 2, 3; Supplementary Fig. 1a–c, rows 1, 2), but it failed to annotate most cells included in the benchmarking analysis (Fig. 2d), potentially leading to information loss. Close examination of the data revealed that AUCell more accurately labeled cell types with a larger number of marker proteins such as ductal cells (Supplementary Fig. 8a, b; Supplementary Data 3), which is in line with AUCell's main focus on scRNA-seq which measures thousands of transcripts but not spatial proteomics technologies which measure tens of proteins.

## AnnoSpat accurately identifies cell types from CODEX data

We further extended our comparative studies to CODEX measurements of 24 proteins in 220,155 cells from 30 islets in a non-diabetic donor (Supplementary Data 9, 10). Similar to IMC results (Fig. 2; Supplementary Fig. 1), qualitative and quantitative studies showed higher performance of AnnoSpat in predicting endocrine cell types with specific antibodies from HPAP CODEX data compared to other supervised algorithms (Supplementary Fig. 9; Supplementary Data 11). SI analysis suggested that AnnoSpat, AUCell, and SCINA were comparable in identifying homogenous cell groups (Supplementary Fig. 9b; Supplementary Data 11). Yet, a close examination of cell labeling revealed that in contrast to AnnoSpat, SCINA-annotated beta cells expressed high levels of somatostatin, a canonical marker of delta cells (Supplementary Fig. 9c, f). While AnnoSpat identified a pure delta cell population, SCINA-annotated delta cells lacked high levels of canonical marker SST. AUCell-annotated delta cells expressed high levels of CD206, ARG1, and CD4, canonical markers of macrophages and helper T cells, respectively (Supplementary Fig. 9c, g). In contrast to IMC analysis, AnnoSpat consistently outperformed SSC in predicting abundant endocrine cells from CODEX data. For instance, ghrelin, a canonical marker of epsilon cells, was highly expressed in SSC-labeled delta cells (Supplementary Fig. 9c, d), supporting the advantage of ELM usage in AnnoSpat. Similar to benchmarking with IMC data (Fig. 2), AnnoSpat outperformed Astir in predicting endocrine cells from CODEX data (Supplementary Fig. 9c, e). Besides beta cells, Astir failed to annotate other major endocrine cell populations (Supplementary Fig. 9e). Close examination of data further showed high levels of non-beta-cell-associated proteins in Astir-labeled beta cells (Supplementary Fig. 9b, e). Notably, we observed high levels of canonical

marker proteins in the nucleus and/or cytoplasm of AnnoSpat-labeled cells from high-resolution CODEX data, further supporting the accuracy of AnnoSpat in cell-type annotation (Supplementary Fig. 9h). Together, these comparative analyses indicated the advantage of using AnnoSpat for accurate, comprehensive, and rapid cell-type annotation from IMC and CODEX spatial proteomic measurements.

## AnnoSpat improves accuracy of cell-type identification in expert-annotated pancreata

To further demonstrate AnnoSpat's ability in accurate cell-type annotation, we compared AnnoSpat- and expert-annotated endocrine cell composition in pancreata of non-diabetic and T1D donors[19] (Fig. 3; Supplementary Data 12). Using Kullback–Leibler (KL) divergence as a measure of difference, we observed concordance in AnnoSpat- and expert-annotated endocrine cell composition in 13 out of 15 (87%) examined IMC samples (Fig. 3a, b; Supplementary Data 13). Notably, AnnoSpat revealed expert cell-type mislabeling in the two discordant samples (Fig. 3g–j). Compared to expert annotation, AnnoSpat identified markedly higher percentages of PP cells in "HPAP002, Head" (Fig. 3a, b). Close examination of staining from "HPAP002, Head" confirmed the accuracy of AnnoSpat's cell-type annotation and showed high expression of the canonical PP-cell marker protein PPY in AnnoSpat-annotated cells (Fig. 3g, h). While AnnoSpat identified a high percentage of alpha cells in the body region of HPAP006 pancreas, expert annotation indicated a low percentage of alpha and a high percentage of delta cells (Fig. 3a, b). In line with AnnoSpat cell-type annotation, we observed a higher percentage of cells with elevated levels of glucagon (a canonical marker of alpha cells) in HPAP006 pancreas body staining (Fig. 3i, j).

Conducting a similar comparative analysis with SSC, Astir, SCINA, and AUCell showed that AnnoSpat outperforms other algorithms in detecting samples with expert cell-type mislabeling (Fig. 3a–f; Supplementary Data 13). KL divergence showed that SSC and SCINA failed to detect expert cell-type mislabeling in "HPAP002, Head". In addition to "HPAP006, Head" that was correctly flagged, Astir-based analysis incorrectly denoted expert cell-type mislabeling in "HPAP014, Body" and "HPAP015, Tail", which was due to Astir's failure to detect PP and epsilon cells in the ROIs from these two donors. AUCell had the lowest accuracy and incorrectly flagged discordance in 14 out of 15 (93%) IMC samples included in the analysis.

Given the single-cell resolution of IMC data, we next used various visualization methods to compare the AnnoSpat-assigned cell types with canonical marker protein expression levels in individual endocrine cells. Uniform manifold approximation and projection (UMAP) plots of AnnoSpat cell label and endocrine marker protein expression clearly visualized specificity of glucagon, C-peptide, somatostatin, ghrelin, and PPY expression in cells labeled as alpha, beta, delta, epsilon, and PP cells, respectively (Supplementary Fig. 8c). A similar analysis using TooManyCells, which visualizes cell–cell protein expression relationships as a tree[13], further confirmed our UMAP analysis and demonstrated a high association between AnnoSpat-predicted endocrine cell types and expression of their canonical marker proteins at cell clusters (Fig. 4a).

Finally, we used the locational information from the spatial proteomic data to directly compare AnnoSpat annotations and marker protein intensities of endocrine cells in situ. This analysis revealed a stark concordance between the position of cells predicted as alpha, beta, delta, epsilon, and PP with the intensity of glucagon, C-peptide, somatostatin, ghrelin, and PPY expression, respectively, on randomly selected IMC and CODEX slides (Fig. 4b, c; Supplementary Fig. 9i, j). This single-cell resolution analysis complemented benchmarking against expert-annotated samples and demonstrated that AnnoSpat exhibited better concordance with expert annotation. Additionally, a thorough examination of IMC data indicated that mislabeling can occur in manual expert annotation, an issue that an accurate automated cell-type annotator could help alleviate.

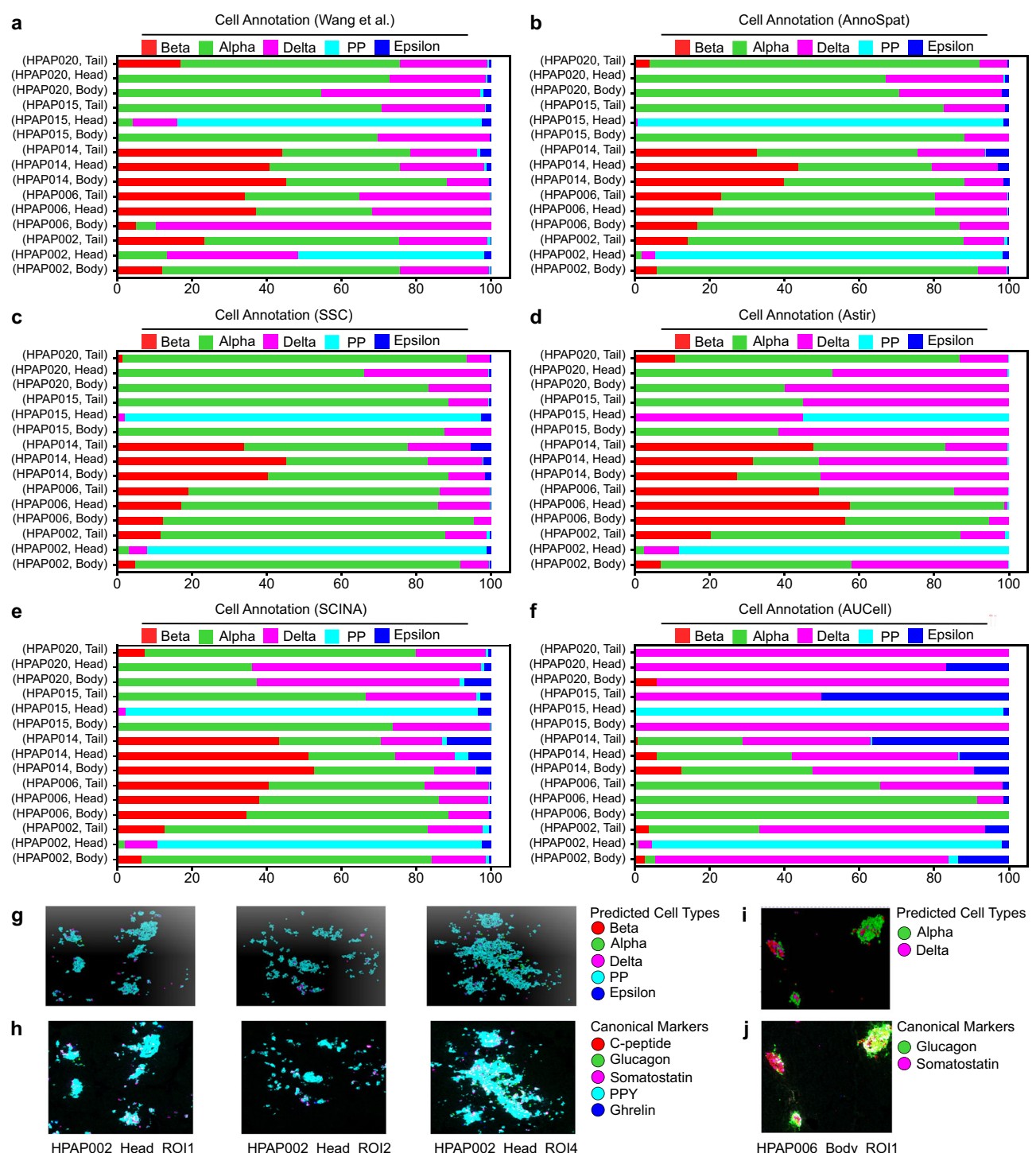

**Fig. 3 | Comparison of expert and automated endocrine cell-type annotation. a–f** Percentage of expert-annotated (**a**), AnnoSpat-annotated (**b**), SSC-annotated (**c**), Astir-annotated (**d**), SCINA-annotated (**e**), and AUCell-annotated (**f**) endocrine cell types from the IMC of different pancreas regions of donors studied in ref. 19. **g–j** Representative IMC images from donors with discordant expert and AnnoSpat cell-type annotation in panels (**a**) and (**b**) are overlaid with AnnoSpat-predicted cell types (**g, i**) or endocrine canonical marker protein channels (**h, j**). C-peptide, glucagon, somatostatin, PPY, and ghrelin marking beta, alpha, delta, PP, and epsilon cells, respectively.

## AnnoSpat's IMC analysis shows an increase in PP-cell count during T1D progression

Linking expression of canonical protein markers with the predicated cell types demonstrated AnnoSpat's ability to automatically and accurately identify various cell types within the heterogeneous pancreas tissue, the site of T1D pathogenesis (Figs. 2, 3; Supplementary Figs. 1, 2, and 7). To further evaluate AnnoSpat's functionality, we next

examined whether it could correctly detect known progressive changes in the pancreata during T1D progression. Thus, we compared IMC data from four non-diabetic (control) and four T1D donors with eight donors with autoantibodies toward islet proteins (AAbs) but without T1D medical history (AAb⁺) (Supplementary Data 2).

Control, AAb⁺, and T1D donors demonstrated distinct total normalized protein expression patterns in cell types annotated by

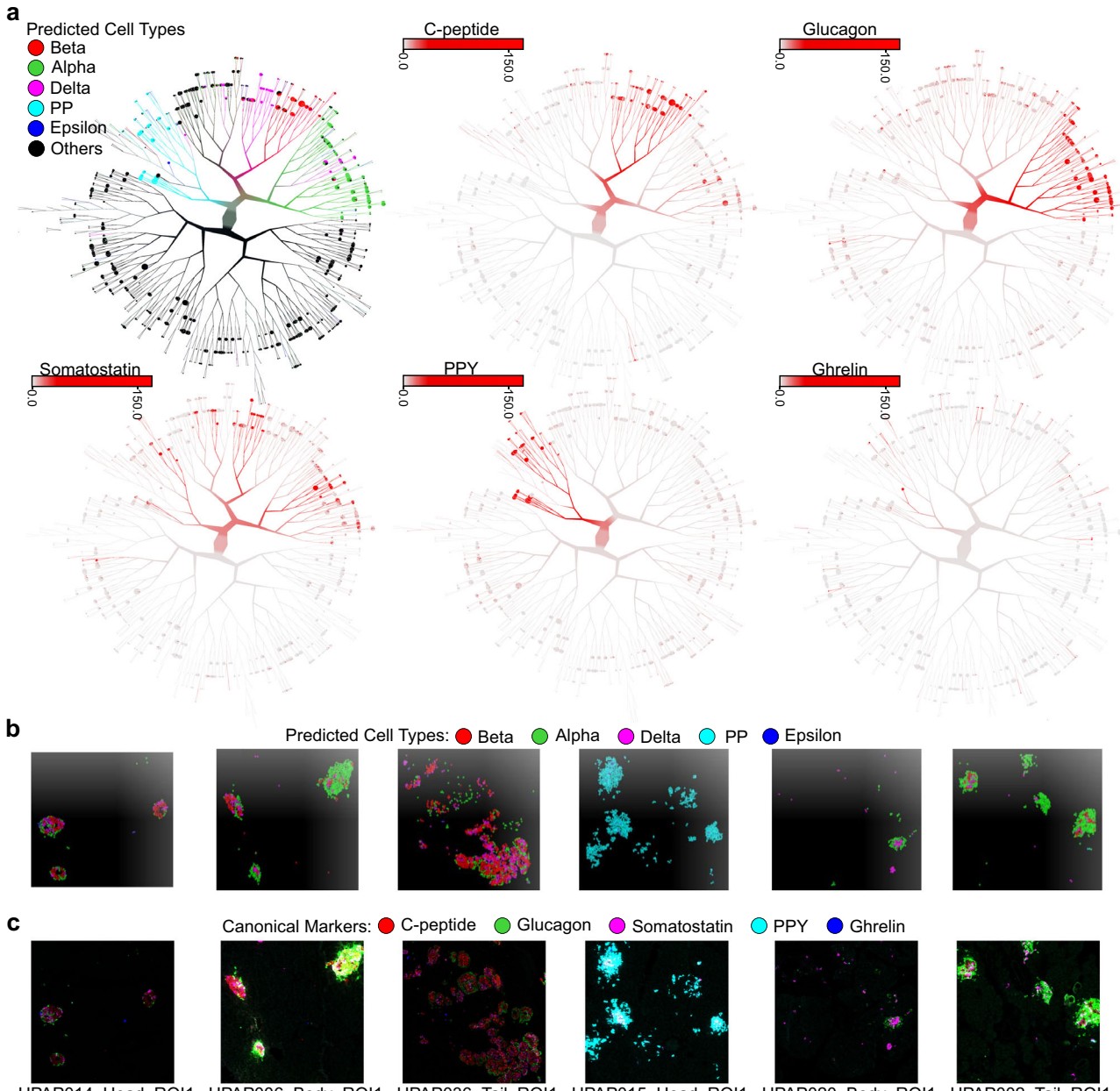

**Fig. 4 | Comparison of protein expression levels and AnnoSpat annotations across pancreatic endocrine cell types. a** From left to right, top to bottom: TooManyCells tree overlaid by AnnoSpat-predicted cell types, and expression levels of C-peptide, glucagon, somatostatin, pancreatic polypeptide protein (PPY), and ghrelin marking beta, alpha, delta, PP, and epsilon cells, respectively in $n = 65,643$ cells across $m = 141$ slides of 16 pancreas donors. **b**, **c** Six representative IMC images from $m = 141$ slides of 16 donors overlaid by AnnoSpat-predicted cell types (**b**) or endocrine canonical marker protein channels (**c**).

AnnoSpat (Fig. 5a). Comparison of cell-type composition revealed marked decreases in beta-cell counts in T1D donors (Fig. 5b), as expected[20,21]. This analysis further indicated a notable increase in the number of cells labeled as PP in T1D donors (Fig. 5b).

　　In contrast to beta cells, the role of PP cells in T1D etiology is less understood. Furthermore, there are conflicting reports regarding changes in the PP-cell count during T1D development[22–27]. We thus compared the number of PP cells identified within the pancreata from control, AAb⁺, and T1D donors. This analysis showed a marked increase in the number of PP cells in T1D pancreata (Fig. 5c), as reported[19,24].

　　To further scrutinize this observation, we examined the location of individual AnnoSpat-annotated endocrine cells (Fig. 5d) on the TooManyCells tree of non-diabetic control and T1D pancreatic cells (Fig. 5e). This single-cell resolution analysis further showed that

AnnoSpat-annotated PP cells were disproportionately located at T1D pancreas heads (Fig. 5e, f), with the exception of HPAP020. Given AAb⁺ donors also did not show elevated PP-cell counts (Fig. 5c), we tested whether disease progression correlates with changes in PP-cell numbers. PP-cell counts were comparable in control and T1D donors with <5 years of T1D, and were markedly lower than donors with a prolonged T1D (Fig. 5g; Supplementary Fig. 10). Notably, fewer PP cells were found in the head of HPAP020 pancreas, a 14-year-old donor who, with missed T1D diagnosis, passed away within days of T1D onset (Fig. 5g; Supplementary Fig. 10). To further substantiate this observation, we closely examined data from Damond et al.[28]. This dataset confirmed our observation and showed enrichment of PP cells in the only donor with long-duration of T1D and an available head section sample in this cohort (nPOD case 6264). Together, these data showed

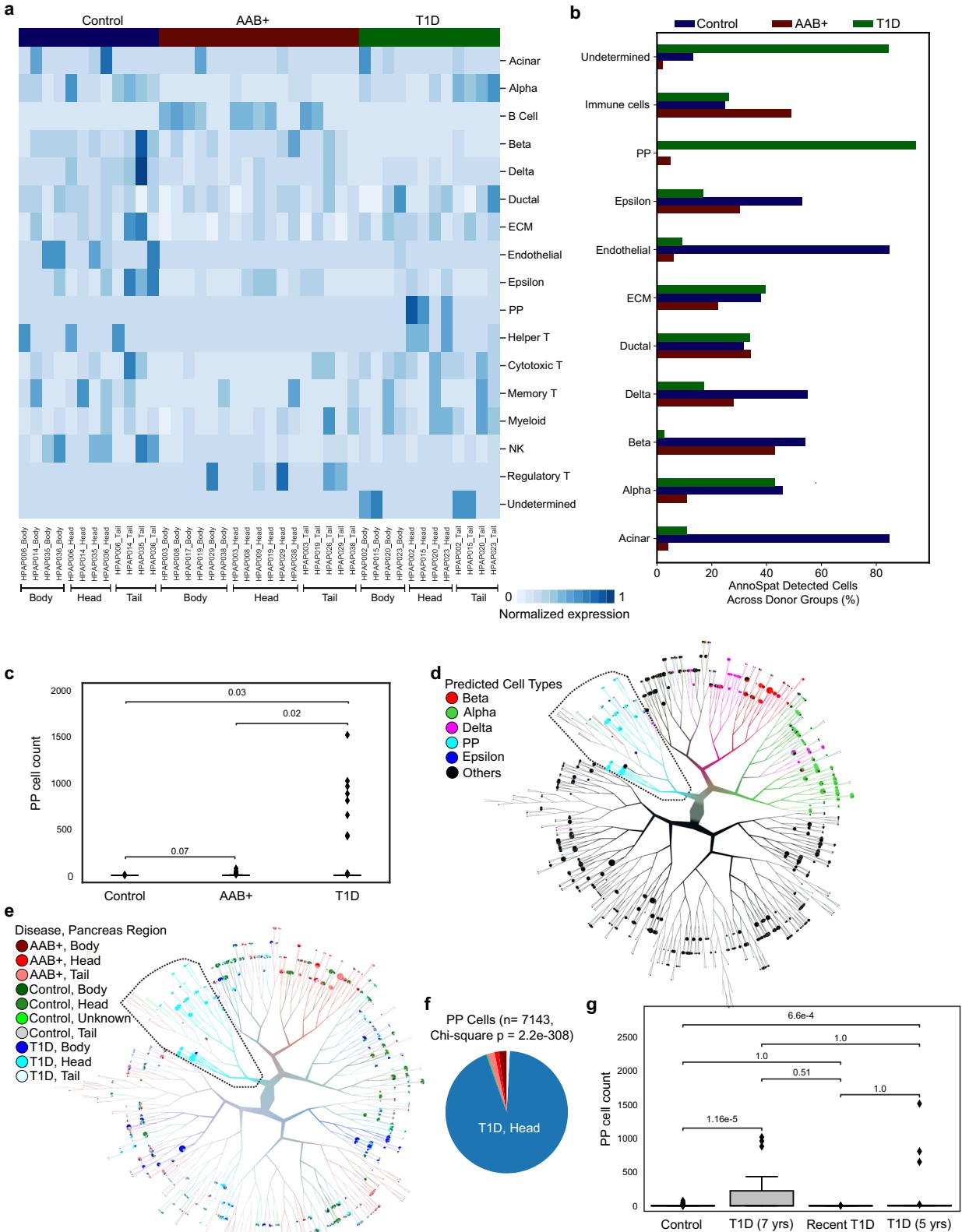

the ability of AnnoSpat to identify rare PP cells, and further suggest changes in the PP-cell count during T1D progression in our cohort, which could be absolute or relative, respectively, due to PP-cell hyperplasia or PP-cell poor region atrophy impacting tissue sampling.

In addition to tissue level analysis (Fig. 5), IMC data can be used for single-cell resolution study of protein expression changes in T1D. To this end, we sought to identify the proliferating cell populations within pancreatic tissue using Ki67 as a protein marker. Average normalized protein levels showed high Ki67 expression in various immune populations (Fig. 6a and "Methods"). To identify the proliferating cell types and their disease status, we used the TooManyCell tree to identify individual Ki67+ cells (Fig. 6b). This analysis revealed that myeloid and regulatory T cells comprised most of the Ki67+ cells (Fig. 6c). Examination of highly proliferating cells' positions further revealed that

**Fig. 5 | PP-cell count increases in the pancreas head during T1D progression.**
**a** Heatmap showing total normalized protein expression for each pancreas region across non-diabetic control, T1D, and AAb⁺ donors. Normalized protein expression for each cell type is calculated by scaling for ROI count per donor pancreas region (3/ROI count) of min–max and TF-IDF normalized expression levels. **b** Bar plots showing the percentage of each AnnoSpat-annotated cell type across pancreata of control, T1D, and AAb⁺ donors. **c** Plots showing PP-cell counts in pancreata from control ($n = 38$), T1D ($n = 46$), and AAb⁺ ($n = 49$) donors. *P* value: two-sided *t*-test with Bonferroni correction. **d**, **e** TooManyCells tree overlaid with AnnoSpat-

predicted cell types (**d**), as well as disease status and pancreas region (**e**). TooManyCells default parameters (quartile normalization and filter threshold of channel intensity <250 and marker protein intensity <1) were used. **f** Pie chart showing fraction of PP cells from different pancreas regions across control, T1D, and AAb⁺ cohorts. **g** Box-and-whisker plots quantifying PP-cell counts in control ($n = 38$) and T1D donors stratified by disease duration (T1D recent $n = 11$, T1D 5 yrs $n = 16$, T1D 7 yrs $n = 19$ ROIs). *P* value: two-sided *t*-test with Bonferroni correction. Box-and-whisker plots: center line, median; box limits, upper (75th) and lower (25th) percentiles; whiskers, 1.5 · interquartile range; points, outliers.

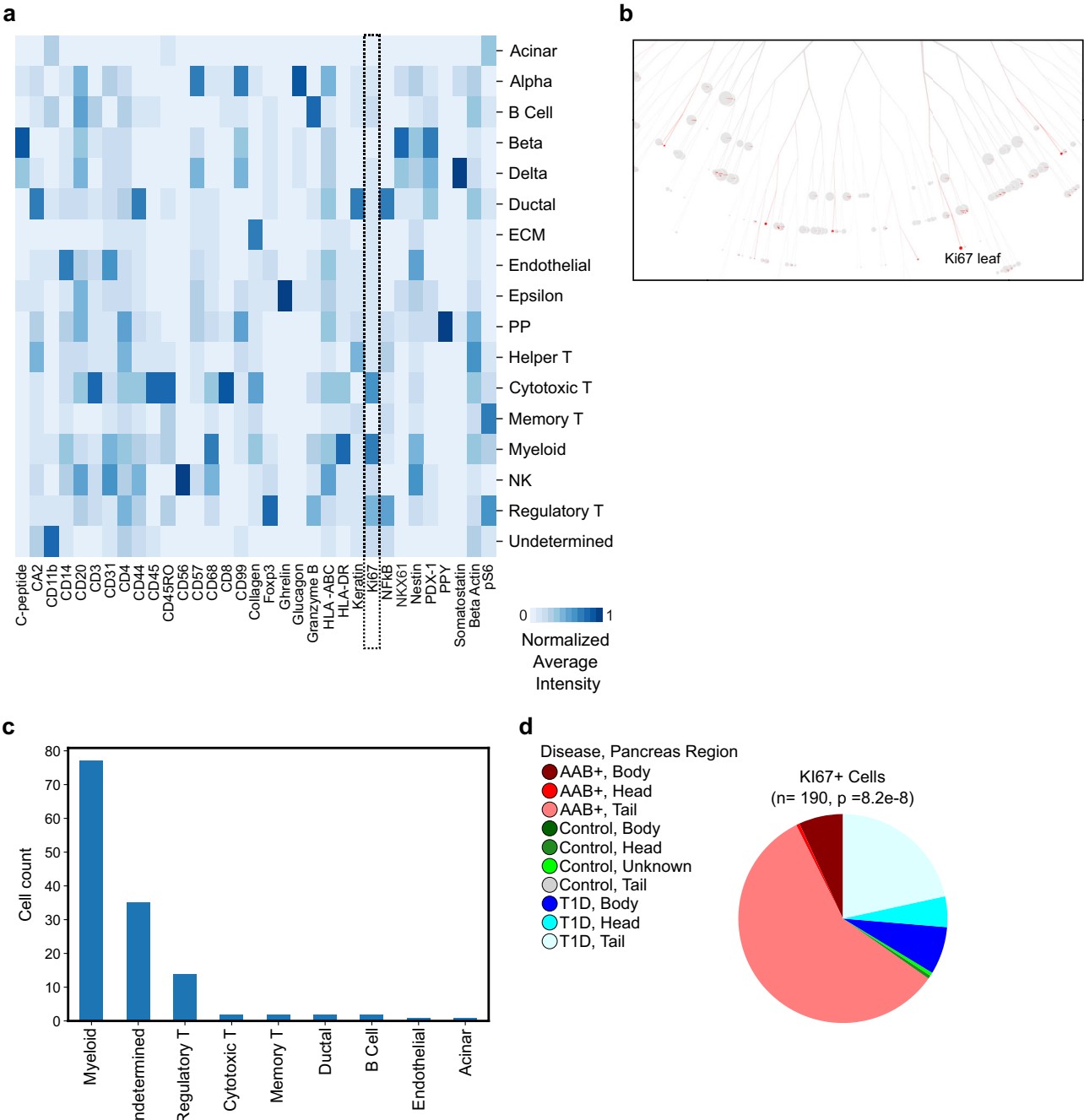

**Fig. 6 | Myeloid and regulatory T cells are hyper-proliferative in T1D pancreata.**
**a** Heatmap showing normalized average expression of all the 33 HPAP IMC-measured proteins across AnnoSpat-annotated cell types. Dash-lined box marking Ki67 column. **b** TooManyCells sub-tree colored by Ki67 expression. **c** Bar plots

showing cell-type count of $n = 190$ cells within the TooManyCells sub-tree in (**b**). **d** Pie chart showing fraction of Ki67⁺ cells from different regions of pancreata of control, T1D, and AAb⁺ donors (*p* value: two-sided chi-square test).

these cells were disproportionately located in the tail region of AAb$^+$ and T1D pancreata (Fig. 6d). Although the role of these highly proliferating immune cells in T1D patients awaits further investigation, this analysis demonstrated the ability of AnnoSpat to simultaneously stratify multiple cell types enabling detailed molecular phenotyping to identify changes in the immune milieu of complex diseases such as T1D.

## AnnoSpat captures known spatial relationship between CD8$^+$ T and islets from IMC data

Having identified the composition of endocrine cells in control, AAb$^+$, and T1D samples, we sought to confirm that AnnoSpat correctly recapitulates known spatial relationships among cell types in pancreata from healthy and T1D donors. To quantify cell proximity, we used AnnoSpat's "Spatial Pattern Finder" functionality, which identifies spatial patterns of cells by reporting cross-correlation functions derived from point process theory. In essence, AnnoSpat treats each cell as a point in space, with the cell type represented as a discrete feature mark. In this space, AnnoSpat measures the expected number of cells per unit area. It then compares this number, which is its null model, to the expected number of cells for a given cell-type pairing to determine whether these cell types tend to aggregate over a range of distances (Fig. 1c and "Methods"). To compare mark cross-correlation functions across different ROIs, we proposed multiple measures for summarizing mark cross-correlation functions into single values. One measure is the distance at the maximum correlation value for each ROI.

To verify the use of mark cross-correlation functions in IMC data, we initially employed AnnoSpat's Spatial Pattern Finder to compare the aggregation of endocrine cell into islets with their aggregation with acinar cells in the ROIs of the control donors. As expected, endocrine cells exhibited a higher degree of aggregation with each other (Fig. 7a, median 2.26 distance at maximum correlation value) than with acinar cells (Fig. 7b, median 149). These spatial relationships were visually confirmed by examining samples at the median values, where endocrine cells tended to aggregate with each other and were positioned more randomly with respect to acinar cells (Fig. 7c, d). Using an alternative measure to summarize the mark cross-correlation functions, we observed similar spatial patterns, thereby confirming the validity of both measures in comparing cell–cell proximity patterns (Supplementary Fig. 11a–d and "Methods").

To further examine the utility of AnnoSpat's Spatial Pattern Finder in studying T1D pathogenesis, we next quantified the spatial relationship between CD8$^+$ T cells and islets. In particular, we focused on quantifying the aggregation of islets with CD8$^+$ T cells, as CD8$^+$ T cells were stained with a more specific antibody in the HPAP IMC panel. Given that the destruction of insulin-producing beta cells by cytotoxic CD8$^+$ T cells contributes to T1D pathogenesis[20,21], we tested the hypothesis that AnnoSpat can accurately recapitulate differential levels of cytotoxic CD8$^+$ T cell infiltration in islets during T1D progression. Applying mark cross-correlation functions to all ROIs for four cohorts – control, AAb$^+$, recent T1D (<1 year), and prolonged T1D (≥1 year) – revealed two distinct patterns of spatial relationships between islets and CD8$^+$ T cells: AAb$^+$ with recent T1D and control with prolonged T1D (Fig. 7e–i). Non-diabetic control donors, as expected[29,30], had relatively low levels of CD8$^+$ T cell infiltration in islets (median 146). Similarly, we observed low levels of CD8$^+$ T cell infiltration in islets of prolonged T1D (median 181), as reported[29,30]. In contrast, both AAb$^+$ (median 81.1) and recent T1D (median 55.7) had markedly higher aggregation of CD8$^+$ T cells within islets relative to both control and prolonged T1D groups (Kruskal–Wallis $p = 5.68e{-}3$) (Fig. 7e–i), recapitulating the expected dynamics of CD8$^+$ T cells aggregation with islets during the natural history of T1D[29–32]. Furthermore, AAb$^+$ and recent T1D tissues showed similar levels of CD8$^+$ T cells aggregation with islets ($p = 0.244$), demonstrating AnnoSpat's ability to detect similar levels of CD8$^+$ T cells infiltration in the early stages of T1D from

IMC data, both with and without clinical diagnosis (Fig. 7e–i). These differential spatial relationships were confirmed using our alternative mark cross-correlation summarization measure (Supplementary Fig. 11e–i). Visual inspection of IMC images supported these quantitative observations (Supplementary Fig. 11j–m), further demonstrating AnnoSpat's ability to recapitulate known changes during T1D progression by accurately quantifying the increase of CD8$^+$ T cell infiltration in islets in early onset but not prolonged T1D.

To demonstrate the advantage of using AnnoSpat for quantifying inter-cell-type spatial relationships, we compared AnnoSpat with HistoCAT[33], which features a similar function to measure cell aggregation. To this end, we repeated the analysis presented in (Fig. 7a–e) with HistoCAT and quantified aggregation of endocrine cells with acinar and CD8$^+$ T cells (Fig. 7j–n). In summary, HistoCAT assesses the grouping between cell type A and cell type B by counting how many cells of type A have neighbors of type B and then dividing that count by the total number of cells of type A with at least one neighbor of type B. To repeat the analysis in (Fig. 7a–e) using HistoCAT, we substituted the distance-dependent mark cross-correlation function ($k(r)$) with HistoCAT's interaction counts (ct) (Fig. 7j–n). HistoCAT and AnnoSpat produced similar spatial relationship results for islets and acinar cells (Fig. 7a, b, j, k), but there were significant differences in the patterns of spatial relationships between CD8$^+$ T cells and islets when quantified by HistoCAT and AnnoSpat (Fig. 7e, n). While AnnoSpat's median mark cross-correlation function significantly differed between control, AAb$^+$, and recent T1D donors, HistoCAT's median interaction counts remained consistent across the three disease conditions (Kruskal–Wallis $p = 0.17$). This suggests that HistoCAT cannot distinguish the expected changes in the aggregation of CD8$^+$ T cells with islets during T1D progression. Notably, HistoCAT analysis failed to discern a significant difference between the levels of CD8$^+$ T cell infiltration within islets in control and recent T1D samples ($p > 0.05$). The marked difference between the spatial analysis outcomes of AnnoSpat and HistoCAT could be attributed to their divergent approaches in quantifying inter-cell-type proximity. HistoCAT uses a fixed neighborhood definition involving the three nearest neighboring cells. By contrast, AnnoSpat considers a range of distances. Together, these spatial aggregation analyses in T1D and healthy donors demonstrate the utility of AnnoSpat in quantifying inter-cell-type proximity relationships.

## Discussion

Spatial profiling of cells in their native tissue environments has enabled comprehensive exploration of cellular organization in tissues. The generation of atlas-scale spatial proteomic datasets has underscored the necessity for automated cell-type annotation methods and tools to quantify cell–cell spatial relationships. However, current methods for cell-type annotation in spatial proteomic analysis often rely on manual labeling, which hinders scalability, or exhibit low accuracy, as demonstrated in our comparative studies. To address this unmet need and to overcome these limitations, we developed AnnoSpat, a solution for the rapid and accurate prediction of individual cell types and quantification of their proximity relationships within spatial proteomic data. Specially, AnnoSpat is tailored to meet the demands of atlas-scale datasets like HPAP, which involve the continuous collection of samples. In contrast to popular unsupervised approaches, AnnoSpat can harness its trainable model to facilitate near-online prediction of cell types in new IMC or CODEX experiments, reducing the need for reanalysis the entire dataset. Using both quantitative and qualitative benchmarking, we demonstrated that AnnoSpat can rapidly and accurately predict the identity of millions of cells in complex human pancreata profiled with IMC and CODEX assays. Our comparative studies further showed that AnnoSpat can predict lineages of large fraction of cells with high accuracy, while other existing cell annotation algorithms failed to do so. AnnoSpat accuracy is further exemplified by identifying endocrine cell populations mislabeled by expert annotation.

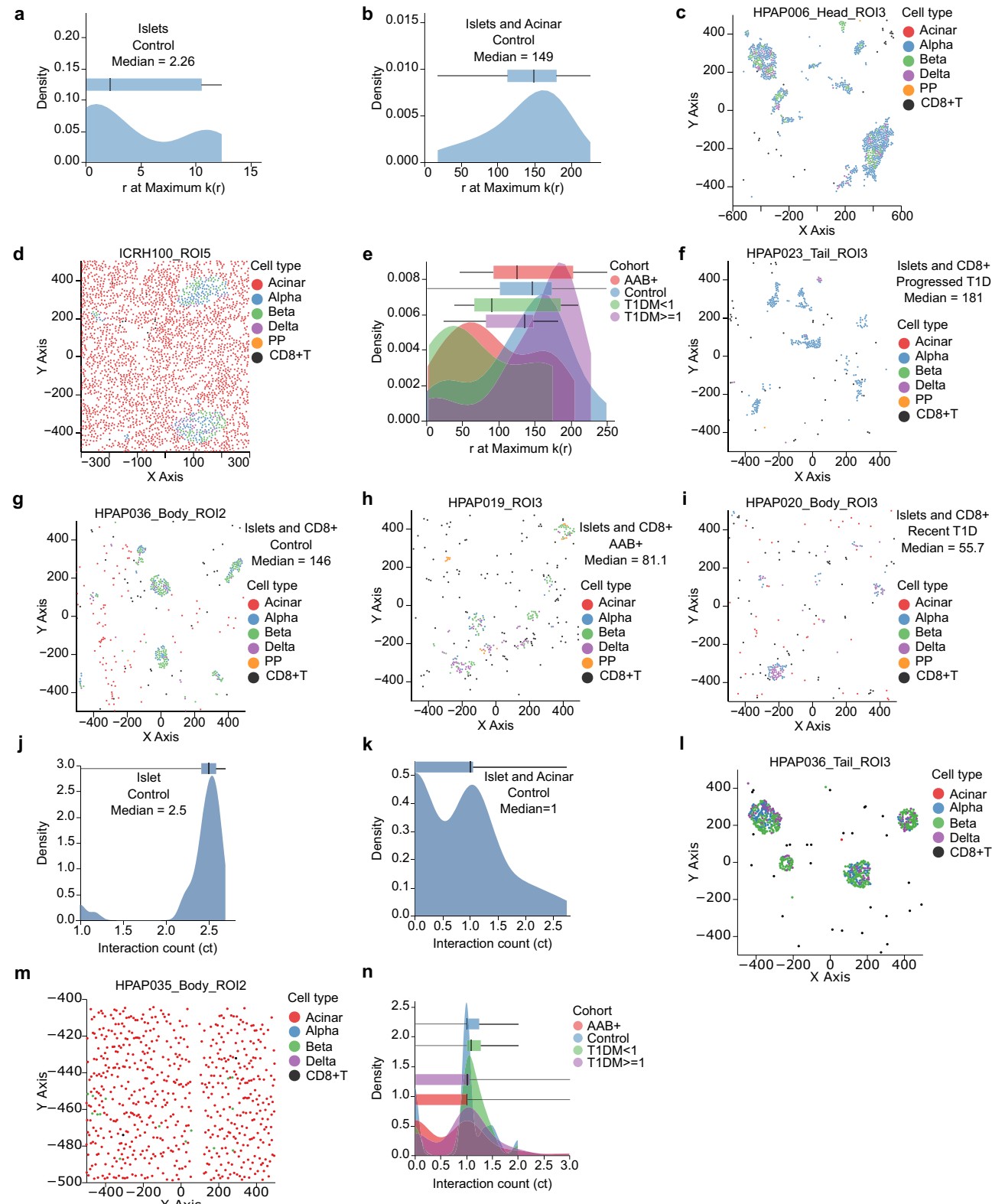

Utilizing AnnoSpat's unique capabilities, we accurately recapitulated known changes in the pancreas microenvironment during the natural history of T1D. AnnoSpat efficiently detected the depletion of beta cells with minimal manual intervention in a dataset of over a million cells. Furthermore, our analysis supported the possibility of changes in the number of PP cells within the pancreas's head region in patients with prolonged T1D. We also observed an enrichment of proliferating immune cells within the tail region of pancreata from AAb⁺ and T1D donors. By employing AnnoSpat's spatial relationship quantification functionality, we faithfully replicated the dynamics of CD8⁺ T cells infiltration within islets during T1D progression. Specifically, AnnoSpat but not HistoCAT successfully detected the expected differential aggregation of CD8⁺ cells in islets when recently diagnosed and AAb⁺ donors were compared with non-diabetic donors and those with prolonged T1D.

**Fig. 7 | The extent of CD8⁺ T cell infiltration in islets changes during T1D progression. a, b** Distribution and box-and-whisker plots of distance $r$ at the $k(r)$ maximum for endocrine cells with respect to themselves (**a**) or acinar cells (**b**) ($n = 48$ ROIs). **c, d** Scatter plots showing cell locations within ROIs at the median of (**a**) and (**b**) distributions at (**c**) and (**d**), respectively. Endocrine cells aggregate more with themselves than acinar cells. **e** Distribution and box-and-whisker plots of the distance at the maximal point in the mark cross-correlation functions across control, AAb⁺, recent T1D, and prolonged T1D showing that greater aggregation of islets with CD8⁺ T cells in AAb⁺ ($n = 49$) and recent T1D ($n = 11$) compared to control ($n = 48$) and prolonged T1D ($n = 35$) ROIs (Kruskal–Wallis Control versus T1DM ≥ 1: $p = 5.68e{-}3$, AAb⁺ versus T1DM < 1: $p = 0.244$). **f–i** Scatter plots showing cell locations within ROIs at the median of each cohort in (**e**). From lowest to highest aggregation: prolonged T1D (**f**), control (**g**), AAb⁺ (**h**), and recent T1D (**i**).

**j, k** Distribution and box-and-whisker plots of HistoCAT interaction counts (ct) for endocrine cells with respect to themselves (**j**) or acinar cells (**k**) ($n = 48$ ROIs). Higher count indicates increased aggregation. **l, m** Scatter plots showing cell locations within ROIs at the median of (**j**) and (**k**) distributions at (**l**) and (**m**), respectively. Endocrine cells aggregate more with themselves than acinar cells. **n** Neighborhood analysis of CD8⁺ T cells and islet cells using HistoCAT cannot differentiate between control and T1D. The distribution and box-and-whisker plots cts across control ($n = 48$), AAb⁺ ($n = 49$), recent T1D ($n = 11$), and prolonged T1D ($n = 35$) do not show higher aggregation of islets with CD8⁺ T in AAb⁺ and recent T1D compared to control and prolonged T1D (Kruskal–Wallis $p = 0.17$). Cells in scatter plots are colored by AnnoSpat-predicted cell types. Box-and-whisker plots: center line, median; box limits, upper (75th) and lower (25th) percentiles; whiskers, 1.5 · interquartile range; points, outliers.

The complexity of pancreatic tissue and the heterogeneity of T1D pathology[30–32,34] underscore the need for algorithms that can enable high-throughput, high-resolution, in situ examination of pancreatic tissue from organ donors with T1D. Recent studies show that insulitis is more frequent shortly after diagnosis. Although the frequency of insulitis (the percentage of islets displaying insulitis) inversely correlates with the duration of T1D, insulitis, and insulin-positive islets can still be found in the pancreata of patients with T1D several years after diagnosis[29–31,34]. These studies further revealed that insulitic lesions can be found in insulin-negative islets, albeit at a much lower frequency compared to insulin-positive islets[30,31]. A recent study of islets from T1D organ donors also found a limited correlation between insulitis frequency and disease duration[30]. Although the relationship between insulitis, microvasculature reorganization, and blood flow changes during T1D progression is not fully understood[35], it is well established that infiltrating immune cells reach the islets through blood, lymphatic vessels, and extracellular matrix spaces, and are influenced by islet microendothelium[29,35]. Here, we demonstrated AnnoSpat's ability to faithfully replicate the dynamics of CD8⁺ T cell aggregation with islets during the natural history of T1D. This example demonstrates the potential utility of AnnoSpat in facilitating future characterizations of T1D heterogeneity and complexity. We postulate that AnnoSpat and other algorithmic advances will pave the way for the integration of spatial proteomics data with readouts from histopathological and microscopic assays. This integration can expand our holistic understanding of T1D pathobiology and facilitate the discovery of links between alterations in islet microvasculature, capillary morphology, endothelium, and immune infiltration with islet dysfunction.

AnnoSpat is a general tool for spatial single-cell proteomic data analysis that has the potential to be applied to various tissue types and disease conditions. However, the performance of AnnoSpat, as well as other automated cell-type annotation algorithms, may be affected by the specificity and sensitivity of IMC and CODEX antibodies, such as those used for PPY and CD4 in the HPAP CODEX and IMC assays, respectively. To improve usability across different domains, AnnoSpat is well documented and can be easily installed as a standalone program through pip at https://github.com/faryabiLab/AnnoSpat. Additionally, we made Annospat's spatial pattern quantification functionality available as part of the `TooManyCells`' suite, which can be found at https://github.com/GregorySchwartz/too-many-cells.

## Methods
### IMC and CODEX data
No new data have been generated in this study. IMC data were obtained from formalin-fixed paraffin-embedded pancreatic tissues collected by the HPAP as described previously[19]. CODEX data were obtained from the same source. Both IMC and CODEX datasets analyzed in this study have been deposited in PANC-DB https://hpap.pmacs.upenn.edu/, data portal developed by the Faryabi Lab for HPAP, and is publicly accessible without any restriction. Relevant clinical data are available both through PANC-DB (https://hpap.pmacs.upenn.edu/)

and Supplementary Data 1–13. In IMC, cell segmentation of all images was performed with the Vis software package (Visiopharm). All image channels were pre-processed with a 3 × 3-pixel median filter. Afterward, cells were segmented by applying a polynomial local linear parameter-based blob filter to the Iridium-193 DNA channel of each image to select objects representing individual nuclei. Identified nuclear objects were restricted to those >10 μm[36]. The detected objects were dilated up to seven pixels to approximate cell boundaries. For all proteins, the average pixel intensity of the channel per cell was exported from Visiopharm and used for AnnoSpat's input. Cell locations on each ROI were also exported for AnnoSpat's input.

### AnnoSpat overview
AnnoSpat is a tool to annotate single cells from their proteomic profiles and measure spatial cellular relationships using their in situ coordinates within the ROI. AnnoSpat takes as input a single-cell raw proteomic data with associated spatial information as well as a Marker Protein file listing potentially both positive and negative protein signatures associated with desired cell types. The format of the Marker Protein file can be found in Supplementary Data 3, 10.

AnnoSpat first normalizes the protein channel intensity data to reduce the effect of outliers and varied protein intensity scales ("Methods: Data processing"). AnnoSpat then randomly splits the normalized data into two partitions (training and testing sets). Cells from 50% of all ROIs are placed in the training set, while the remaining are used as the testing set. If the ROIs' disease condition/status is available, AnnoSpat can stratify the ROI split by disease status to ensure that an equal percentage of each disease status is included in each of the training and test sets.

AnnoSpat can use the cell-type labels and cellular coordinates to quantify spatial relationships between each pair of cell types ("Methods: AnnoSpat's Spatial Pattern Finder"). Briefly, AnnoSpats use point process theory to quantify relationships (aggregation or repulsion) between any two cell types across a range of distances. This information is summarized with a variety of different metrics including the distance at the maximum correlation, the distance at which the correlation first becomes positive or negative, and more in order to quantify proximity relationships across ROIs. Interactive plots of each cell location with observed feature (protein expression) distributions are also outputted to facilitate data exploration (e.g., see Fig. 7; Supplementary Fig. 11).

AnnoSpat implements constrained K-means semi-supervised clustering[37] to identify groups of cells in the training set that are similar in proteomic space. AnnoSpat's constrained K-means clustering is initialized by "initial cluster centroids", providing cell-type aware clustering ("Methods: Generation of initial cluster centroids"). The number of clusters is deterministic and is equal to $K + 1$, where $K$ denotes the number of expected cell types in the sample. The additional $(K + 1)$th cluster accounts for other cell types in the experiment that are not specified in the Marker Protein file, including Unknown ones. The output of constrained K-means produces the cells that are predicted to be related and thus are used by AnnoSpat as a training set to learn the label of the remaining cells by training an ELM classifier[12]

("Methods: Training extreme learning machine classifier"). The trained model is saved to label cells from other data sources, eliminating the need for re-clustering or re-training whenever new data are available.

## AnnoSpat data processing

To reduce the effect of outliers, AnnoSpat first calculates Data matrix $D$ by log transforming cell-by-protein channel intensity (expression) after the addition of pseudo-count 1. Specifically, $d_{c \times p} = e_{c \times p} + 1$, where $e_{c \times p}$ is the expression of protein p channel in cell c. Then, AnnoSpat unit normalizes the log-transformed intensity matrix to scale each cell vector to unit length. This projects each cell to a unit sphere in the proteomic space. We denote the normalized proteomic matrix by $X$ obtained from scaling each row $d_i$ of $D$ as follows:

$$x_i = \frac{d_i}{||d_i||}, \text{ where } ||d_i|| = \left( \sum_{j=1}^{P} ||d_j||^2 \right)^{1/2}, \qquad (1)$$

where $||d_i||$ denotes the $l_2$ or Euclidean norm of $i$th cell. $P$ is the number of measured proteins.

This step accounts for variable expression across proteins and correlates the Euclidean distances (used for clustering) between cell vectors and cosine distances in the proteomic space. Compared to Euclidean distance, the angle between the cell vectors in proteomic space better reflects cell–cell similarities/differences[38].

## Generation of initial cluster centroids

As opposed to traditional K-means where the initial cluster centroids are randomly selected, AnnoSpat implements constrained K-means that follow a more cell-type aware approach[37]. Initial cluster centroids are obtained from representatives of each cluster (cell type here). AnnoSpat calculates initial cluster centroids by taking the mean of representative cells $R_k$ for each cluster $k = 1, \ldots, K+1$. The number of clusters is one more than the number of cell types $K$; an extra (Unknown) cluster accounts for cell types not included in the Marker Protein file.

AnnoSpat obtains the cluster representations $R_1, R_2, \ldots, R_K$ by:

1. Obtaining positive and negative markers $M^+$ and $M^-$ from the Marker Protein file.
2. Calculating the score $M_c$ for $c$th cell type by multiplying the protein intensities corresponding to positive markers and the compliment of protein intensities corresponding to negative markers as follows:

$$M_c = \prod_{i \in M^+} X_i * \prod_{j \in M^-} (\max(X_j) - X_j), \quad c = 1, \ldots, K. \qquad (2)$$

3. Selecting cell representatives $R_1, R_2, \ldots, R_K$ of cell types $c = 1, \ldots, K$ in the Marker Protein file such that they have

$$M_c > M_{c,\text{high}} \quad \text{and} \quad M_c < M_{c,\text{max}}, \text{where} \qquad (3)$$

$M_{c,\text{high}} = \text{percentile}(M_c, q_{\text{high}})$ and $M_{c,\text{max}} = \text{percentile}(M_c, q_{\text{max}})$.

The value $q_{\text{high}}$ is adaptive and can be optionally chosen based on prior knowledge of the number of cells from the cell type present in the data (defaulting to the 95th percentile). Here, $q_{\text{high}}$ was set to $99 \leq q_{\text{high}} \leq 99.9$ and $99.5 \leq q_{\text{high}} \leq 99.99$ for various cell types in the analysis of pancreas IMC and CODEX data, respectively. $M_{c,\text{high}}$ is the score cut-off to pick cluster representative cells as the ones having a very high score $M_c$ corresponding to the $c$th cell type. The threshold $q_{\text{max}}$ is set to 100 or a value slightly less than that to make sure that assay artifacts are not included in the initial cluster centroid calculation. Here, $q_{\text{max}}$ was set to 99.999 and 100 for the analysis of pancreas IMC and CODEX data, respectively.

$M^+$ and $M^-$ cannot be defined for the Unknown cluster, since multiple cell types could be captured in this cluster. Hence, after calculation of $M_c$ for cell types $c = 1, \ldots, K$ with specified markers, the intersection of all cells with low $M_c$ value for all cell types $c = 1, \ldots, K$ are identified. The intuition here is that cell types whose markers are not specified in the Marker Protein file (i.e., Unknown cells) should not be representatives of any cell type $c = 1, \ldots, K$, and thus $M_c$ score should be low for the Unknown cells with respect to all the cell types $c = 1, \ldots, K$ with known markers.

In other words:

1. For each cell type $c = 1, \ldots, K$,
   (a) Calculate $M_c$ (using $M^+$ and $M^-$ of cell types defined in the Marker Protein file)
   (b) Obtain the cells with $M_c < M_{c,\text{low}}$ where

$$M_{c,\text{low}} = \text{percentile}(M_c, q_{\text{low}}). \qquad (4)$$

The threshold $q_{\text{low}}$ defines the cut-off to choose cells with expression $< M_{c,\text{low}}$ in cell types $c$.

This will pick cells $U_c$ that belong to cell type $c$ with a very low probability.

2. The Unknown class is identified by taking the intersection of $U_1, \ldots, U_K$ sets. These cells are least likely to represent any of the cell types defined in the Marker Protein file.

AnnoSpat performs the above procedure to assign cluster representative cells in decreasing order of cell-type abundance (representative of more abundant cell types are selected first). Cell-type abundance acts as a proxy for the expected number of cells for each cell type and is obtained by summing cell intensities of the scale-normalized canonical protein markers.

Once the cell representatives $R_1, R_2, \ldots, R_{K+1}$ have been assigned, AnnoSpat computes initial centroids $\bar{x}_k$ for cluster $k = 1, \ldots, K+1$ by taking the average across the representative cells $R_k$ as follows:

$$\bar{x}_{kj} = \frac{1}{|R_k|} \sum_{x_i \in R_k} x_{ij}, \text{ for } j = 1, \ldots, P \qquad (5)$$

where $x_{ij}$ represents the intensity of the $j$th protein in $i$th cell.

## Cell labeling with semi-supervised clustering

AnnoSpat takes the cell representatives $R_k$'s and initial cluster centroids $\bar{x}_k$'s and iteratively runs a constrained K-means algorithm on the cells from 50% of the ROIs included in the training set as shown in Algorithm 1. $L_i$ denotes the cluster label assigned to the $i$th cell and $C_k$ denotes the set of cells in cluster $k$. The assigned cell labels are the predicted cell types of training data for the AnnoSpat's Annotator.

**Algorithm 1. Constrained K-means**
 **Initialize** $K$, $n$
 **Input**: normalized data $X$, initial centroids $\{\bar{x}_1, \ldots \bar{x}_{K+1}\}$ and cell
 representatives $\{R_1, R_2, \ldots, R_{K+1}\}$
 **For** $iter = 1, 2, \ldots, n$
 Cluster assignment:
 When $x_i \in R_k$, $L_i = k$
 Otherwise, $L_i = \arg \min_k ||x_i - \bar{x}_k||^2$
 Centroid computation:
 $\bar{x}_k = \frac{1}{|C_k|} \sum_{x_i \in C_k} x_i$
 **End for**
 **Return**: Labels $L$, centroids $\bar{x}$

## Training extreme learning machine classifier

AnnoSpat uses the cell-type labels $L$ predicted by its semi-supervised clustering algorithm as training labels $Y_{TR}$ to then learn an ELM classifier[12]. The classifier predicts the label of remaining cells in new

ROIs not included in the training data. We implemented ELM in AnnoSpat because it is a single-layer feed-forward neural network classifier and does not need to be iteratively tuned via backpropagation. This would enable AnnoSpat to learn accurate cell-type prediction models markedly faster than gradient-based learning techniques.

Comparative analysis during the design of AnnoSpat confirmed earlier studies[12] and showed that, while ELM and SVM provide comparable accuracy in annotating ~1,170,000 cells in our IMC dataset (Fig. 2c, h), ELM was ~2 times faster than SVM (73.9 versus 159.9 s). These characteristics make it a more suitable classifier for near-online annotation of atlas-scale datasets, including HPAP.

AnnoSpat's ELM is implemented as follows:

1. Assign input layer weights $W_I$ and bias $b_I$ randomly from normal distributions:

$$W_I \sim \mathbb{N}(0, \mathbb{I}) \tag{6}$$

$$b_I \sim [N(0,1)] \tag{7}$$

2. Compute hidden layer output $H$:

$$H = \phi(W_I * X_{TR} + b_I) \tag{8}$$

Here, $\phi$ denotes the activation function used at the hidden layer, and $X_{TR}$ is the normalized protein intensity of the training set.

3. Compute the output layer weights $W_O$

$$W_O = H^{\dagger} * Y_{TR} \tag{9}$$

Here $H^{\dagger}$ is the Moore–Penrose inverse of hidden layer output matrix H. The training labels $Y_{TR}$ are transformed into a one-hot encoded format to avoid ordinal relationship interpretability between cell types by the model.

Once the output weights are learned, the types (labels) of new cells $Y_{TS}$ can be predicted from their normalized protein expression $X_{TS}$ by the learned weights in ELM:

$$Y_{TS} = \phi(W_I * X_{TS} + b_I) * W_O \tag{10}$$

To demonstrate AnnoSpat's generalizability, we additionally used the trained AnnoSpat model to annotate cells in two additional ROIs from HPAP organ donors that were not part of the original 143 slides of 19 donors and examined the accuracy of cell-type annotation.

## Cell-type prediction with unsupervised clustering algorithms

To predict cell types with unsupervised clustering, we used K-means, Seurat[16], FlowSOM[18], and PhenoGraph[17] clustering followed by differential protein expression analysis between cells in each cluster versus cells in all other clusters.

K-means clustering was implemented using `scikit-learn` library with default parameters. Similar to AnnoSpat, the number of clusters for K-means clustering `n_clusters` was set to 17.

`CreateSeuratObject` function from Seurat R package was used to process the protein expression table. `NormalizeData` and `ScaleData` functions were then used to for data normalized and scaled. `RunPCA` function was used to perform principal component analysis on the normalized counts. Given the number of cell types that could be detected by the HPAP IMC panel, we used `FindNeighbors` function with dims = 1:30, and `FindClusters` function with resolution equal to 0.3 (T1D), 0.2 (Control), and 0.2 (Combined), which resulted in 17, 19, and 22 clusters in T1D, Control, and Combined cohorts, respectively.

To cluster cells with FlowSOM, first, `read.flowSet` function from openCyto R package was used to convert CSV files of raw protein expression values to FCS objects (note: this function is not exposed in the package's API). Then, `FlowSOM`, `BuildSOM`, and `BuiltMST` functions from FlowSOM R package were used to construct a self-organizing map (SOM) and minimum spanning tree (MST). Given the number of cell types that could be detected by the HPAP IMC panel, we finally used `metaClustering_consensus` function with `k` equal to 17 to perform meta-clustering on MST.

Phenograph expects normalized expression values. Hence, `NormalizeData` function from Seurat R package was used to normalize raw expression values. `pandas.read_csv` method from Pandas dataframe was used to input normalized protein expression CSV files. Given the number of cell types that could be detected by the HPAP IMC panel, we used `phenograph.cluster` function with `k` equal to 1000 (T1D), 500 (Control), and 1000 (Combined), which resulted in 21, 23, and 37 clusters in T1D, Control, and Combined cohorts, respectively.

The out of each clustering method was used for a series of one-versus-all-others differential protein expression analyses. We used Mann–Whitney $U$ test to determine the significance of differences in protein expression levels for cells in each cluster versus cells in all other clusters. The log2 fold change (FC) of each cluster's mean expression of a given protein was used as a further measure of the difference between cell clusters. The final measure of significance for each cluster was calculated as $-\log_{10}(p - value) \times \log_2 FC$. To assign cell-type labels, the most deferentially expressed protein for each cluster was queried against the Marker Protein file in Supplementary Data 3 for cell-type assignment. For all clustering algorithms, the order of cell-type label assignment was the same as the order of columns in Supplementary Data 3.

## Comparison of algorithm-annotated and expert-annotated cell types

Endocrine composition of expert-annotated cell types in different sections of pancreata from five HPAP donors was obtained from ref. 19. Wang et al. borrowed approaches commonly used in flow cytometry data analysis to manually annotate the cells in each individual ROI[19]. In summary, the cells were annotated by a combination of manual gating and thresholding based on Gaussian mixture models that were used to separately identify positive and negative cutoffs for each individual protein channel in each ROI, followed by so-called Boolean rules. These rules start assigning the cells to alpha, beta, delta, PP, and epsilon, followed by other cell types with available markers in HPAP IMC assay, in that order, based on "Positive" and "Exclusion" markers as listed in ref. 19. For each (donor, section), we used minimum KL divergence to determine distance between the Wang et al. reported endocrine cell-type distribution and AnnoSpat-, SSC-, Astir-, SCINA-, or AUCell-annotated endocrine cell-type distribution. Two endocrine cell-type distributions were called discordant if their minimum KL divergence was >0.4.

## Data processing for visualization

Data for heatmaps in Fig. 2, as well as Supplementary Figs. 2c–g, 3, 4a–c, 5a, b, 6a–e, and 7 have been normalized to penalize the expression of non-specific proteins using an analog variant of TF-IDF normalization after min–max scaling of protein expression. The specificity of a protein can be quantified as an inverse function of the number of cell types in which it is expressed (its abundance across various cell types). Hence, the normalized value of protein $p_i$ is obtained by multiplying each value by the logarithm of ratio of total protein abundance $p_{total}$ in the data and the abundance of that protein across all cell types $p_{sum}$. If $p$ is the expression of protein $p_i$ in cell $c_j$, then the normalized value is calculated by:

$$p_{i,j}^{TF,IDF} = p * \log\left(\frac{p_{total}}{p_{sum}}\right). \tag{11}$$

In min−max normalization, min and max values are the 0.01th and 99.99th percentile expressions, respectively.

**AnnoSpat's Spatial Pattern Finder: quantification of cell proximity pattern**

In order to quantify the relationships between cell types in the T1D pancreas, we interpreted the cell locations and cell-type labels as a marked point pattern. A point pattern provides the locations of observations; here, cell locations are represented as Cartesian coordinates. Each cell can have additional features known as marks; here, each cell's mark is the predicted cell type. By realizing the marked point pattern as a random marked point process, we can quantify cell-type spatial relationships. A point process is a random set of points, where the number of points and their locations are both random. Using point process theory, we can understand the relationship between cell types not as a single index, but rather as many values resulting in the formulation of a given function of distance $r$.

The standard model of a point process $\gtrsim$ assumes that the process extends all space, but the observed region is bounded by a window $W$. Then we can define the data as an unordered set[39]

$$\psi = \psi_1, \ldots, \psi_n, \psi_i \in W, n > 0, \tag{12}$$

the point pattern of $\boldsymbol{\Psi}$.

Now we can define our ROI within the context of marks. Consider the marked point pattern as an unordered set of cells observed within a window $W$ with marks in $M$,

$$\gamma = (\psi_1, m_1), \ldots, (\psi_n, m_n), \psi_i \in W, m_i \in M, \tag{13}$$

where $\psi_i$ is the location and $m_i$ is the mark of cell $i$, respectively[39]. Marks may be continuous real numbers, such as cell size, or discrete, such as cell type. Our objective is to quantify the dependence between the marks of two cells of distance $r$ apart in the marked point process $\boldsymbol{\Gamma}$. This dependence, known as the mark correlation function $k_f(r)$, is informally defined as[39,40]

$$k_f(r) = \frac{\mathbb{E}_{i,j}[f(M_i, M_j)]}{\mathbb{E}[f(M, M')]}, \tag{14}$$

where $M_i$, $M_j$ are marks of two cells separated by distance $r$, $M, M'$ are independent realizations of the marginal distribution of marks, and $\mathbb{E}$ is the intensity of a point process, or the average density of points (the expected number of points per unit area), and where $\mathbb{E}_{i,j}$ is the conditional expectation that there exist cells at locations $i$ and $j$ separated by distance $r$. While $f$ is any function that returns a non-negative real value, we commonly use $f(m_1, m_2) = m_1 m_2$ for continuous marks and $f(m_1, m_2) = \mathbb{1}(m_1, m_2) = 1$ where $m_1 = m_2$ and $= 0$ for everything else for discrete (categorical) marks[39]. Then, $k_f(r) = 1$ suggests a lack of correlation such that under random mark labeling, $k_f(r) \equiv 1$. The interpretation of greater than or less than 1 would be determined by the chosen function $f$, but throughout this study, we interpret >1 as correlated and <1 as anti-correlated. This mark correlation function, however, assumes that cell type would be a single mark and does not specify the relationship between, for instance, CD8+ T cells and islet cells.

To understand the relationship between any two cell types, we expand the mark correlation function $k_f(r)$ to define the mark cross-correlation function, $k_{mm}(r)$. Here, instead of $m_i \in M$ as a single mark, we define $\mathsf{m}_{ia} \in M$ as the value of mark $a$ in cell $i$ from the row vector of marks $\mathsf{m}_i$ attached to cell $i$. Instead of a single mark for cell type, we convert the mark into a mark row vector $m_i$ for cell $i$ containing $c$ entries, where each index $0 < j \le c$ represents an indicator value for cell type $a$. In short, $\mathsf{m}_{ia} = 1$ indicates that the cell $i$ is of cell type $a$.

Using this expanded mark vector, we can define the mark cross-correlation function[39] as

$$k_{mm}(r) = \frac{\mathbb{E}_{i,j}[f(M_{ia} M_{jb})]}{\mathbb{E}[f(M_a, M_b)]}, \tag{15}$$

where $M_{ia}$ and $M_{jb}$ are the marks $a$ and $b$ attached to cells $i$ and $j$, respectively, while $M_a$ and $M_b$ are independent random values drawn from all cells at mark indices $a$ and $b$, respectively. Here, $f$ is defined as with the mark correlation function. Using categorical marks for cell types, we then interpret $k_{mm}(r) > 1$ as correlated, $<1$ as anti-correlated, and $= 1$ as random. We carried out all mark cross-correlation analyses using the spatstat R package[39].

The output of each mark cross-correlation function on an ROI is a series of correlation values as a function of distance $r$. To compare across several ROIs, we summarized each curve by either the $r$ at the maximum $k_{mm}(r)$ ($\max_r k_{mm}(r)$) (Fig. 7) or the log-transformed ratio of the maximum $k_{mm}(r)$ to the $r$ at the maximum $k_{mm}(r)$ ($\log \frac{\max_r k_{mm}(r)}{\arg\max_r k_{mm}(r)}$) (Supplementary Fig. 11). The former value decreases with increasing aggregation (the highest correlation is with cells with smaller $r$) while the latter increases with increasing aggregation. To compare distributions, we used Kruskal−Wallis one-way analysis of variance for multiple hypotheses followed by pairwise Mann−Whitney $U$ tests.

### Statistics and reproducibility

Statistical tests used in data analysis are listed in figure legends and/or relevant sections in "Methods". No statistical method was used to predetermine the sample size. No data were excluded from the analyses. The experiments were not randomized.

### Reporting summary

Further information on research design is available in the Nature Portfolio Reporting Summary linked to this article.

## Data availability

No new data have been generated in this study. The IMC and CODEX datasets analyzed in this study have been deposited in PANC-DB https://hpap.pmacs.upenn.edu/, the data portal of Human Pancreas Analysis Program (HPAP) consortium (RRID: SCR_016202) developed by the Faryabi Lab, and is publicly accessible without any restriction. Relevant clinical data are available both through PANC-DB (https://hpap.pmacs.upenn.edu/) and Supplementary Data 2. Processed IMC data are provided as a Source Data file.

## Code availability

AnnoSpat is available at https://github.com/faryabiLab/AnnoSpat. Spatial Pattern Finder is also available as part of the TooManyCells suite located at https://github.com/faryabib/too-many-cells.

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

## Acknowledgements

This work was supported in part by R01-CA230800, R01-CA248041 (to R.B.F.), Canada Research Chairs Program, Canadian Cancer Society Challenge Grant 707484 (to G.W.S.), Human Islet Research Network (RRID: SCR-014393), and Human Pancreas Analysis Program (RRID: SCR-016202) through DK112217 (to A.N., K.H.K.), DK123594 (to A.N., K.H.K.), DK104211 (A.C.P.), DK108120 (A.C.P.), DK112232 (A.C.P.), DK123716 (A.C.P, R.B.F., M.B.), and DK106755 (A.C.P.).

## Author contributions

Conceptualization: R.B.F., G.W.S.; methodology: A.M., G.W.S., R.B.F.; software: A.M., G.W.S.; investigation: A.M., G.W.S., R.B.F.; formal analysis: A.M.,D.T., D.S., G.W.S., R.B.F.,F.T.Z, Y.Z, N.G.B.; resources and reagents: R.B.F., Y.J.W., A.C.P., K.H.K., G.V., A.N.; writing–review & editing: G.W.S., R.B.F.; writing–original draft: A.M., G.W.S., R.B.F.; supervision: R.B.F.; funding acquisition: R.B.F., G.V., A.N.

## Competing interests

The authors declare no competing interests.
