## [Peer Review File · Nature Communications]

Reviewers' Comments:

Reviewer #1:

Remarks to the Author:

In this manuscript, Mongia et al. have developed AnnoSpat which identifies cell types and performs cell-cell proximity analysis using spatial proteomics data obtained from IMC and CODEX techniques. The authors combined semi-supervised and supervised learning methods for cell type annotation on IMC data from pancreatic tissues, and they compared AnnoSpat's performance with other methods such as SSC, SCINA, AUCCell, and expert-annotated results. For the spatial analysis, the authors applied point process theory to model cell-cell relationships at different distances. And they found the spatial patterns between CD8+ T cell and islets during the progression of type 1 diabetes (T1D). However, there are some concerns that need to be addressed by the authors as follows:

1. If we believe that semi-supervised constrained K-means clustering results are reliable and use them as the training dataset, why not employ this strategy for the entire dataset? The authors should explain the rationale behind why it is necessary to perform the cell annotation using the 2-stage approach, and how does it bring additional advantage over the single-stage approach in SSC?

2. The authors should provide explain why using ELM in the second-stage cell type classification because it looks very random. How important is this selection? Note that speed is not the ultimate concern in terms of cell annotation tasks. In addition, there are many other classifiers available which don't perform backpropagation.

3. The authors should also include unsupervised clustering strategies in their performance comparison rather than constraining the comparison between supervised methods without training data. Ultimately, the goal of cell type annotation is to identify individual cell types, and unsupervised methods are still the dominant methods in the literature for this goal. So it is important to prove the proposed method provides benefits over existing methods.

4. In the result section "AnnoSpat accurately identifies cell types in complex pancreatic tissues", there are some immune cell types in the T1D dataset but why the authors only compared the annotation results for endocrine cells (from Line 119 to 122). Importantly, it should include in the evaluation more challenging cell types, e.g., cell types that can somehow overlapped such as CD4 T vs T reg, or cell types that show only subtle differences, e.g., effective T vs memory T, etc.

5. It is not very straightforward to visually compare the performance between different methods using the heatmaps showing marker proteins for endocrine cells (e.g. Figure 2A, B, and C). It could be better if the authors can quantify the results.

6. In the result section "AnnoSpat improves the accuracy of cell type identification in expert-annotated pancreata", how were the expert-annotated cells obtained? Were the cells annotated by experts manually or from the original paper based on a set of Boolean rules?

7. The mark cross-correlation function across ROIs is a method for systematic quantification for all images and also an appropriate visualization method to compare cell-cell relationships between different groups. However, the spatial patterns from AnnoSpat are the cell-cell aggregation or dispersion, which is similar to neighborhood analysis (HistoCAT, <https://www.nature.com/articles/nmeth.4391>, R implementation: <https://github.com/BodenmillerGroup/neighborhood>),. So I am wondering how AnnoSpat gains additional meaningful cell-cell relationships that cannot be found in neighborhood analysis from HistoCAT?

Reviewer #2:

Remarks to the Author:

The manuscript by Mongia et al. titled "AnnoSpat annotates cell types and quantifies cellular arrangements from spatial proteomics" presents a method for annotation and quantitative localisation of cell types in intact tissues. While the authors present seemingly successful and interesting results, the methodology itself includes many opaque parameters and appears dependent on good quality prior information, suggesting that it may not generalise well to other applications outside of those presented. More detailed comments can be found below.

It seems to me that the success of this method is extremely dependent on the initial clustering, but there is limited results and discussion on the effect of poor (or sub-optimal) initialisation. For example, if one cell-type is omitted from the initialisation, how does this affect the classification of the remaining cell-types? Conversely, if a cell-type is included in the initialisation, that is not present in the data, how does this affect the classification of the remaining cell-types?

It is also not clear how the user should select the quantile q_{high} for each cell type. This appears to be arbitrarily selected – can the authors comment on the effect setting this parameter? What happens if this is set too low? (what effect does excluding data/introducing a bias in the representative cells have on the classification) And too high? (what effect does including 'outliers' have on the classification)

How is M_c calculated for the "unknown" class? If it is truly unknown, then equation on line 473 cannot be used, as M_+ and M_- is not known (and indeed could be contradictory if multiple cell-types are captured in "unknown").

How was concordance measured for Figure 3A vs 3B? It is not clear how the authors concluded that there is 80% agreement between expert and AnnoSpat. How do the other methods evaluated (SSC, Astir, SCINA, AUCell) perform when compared to the expert annotations?

Why were 50% of the cells from each ROI included in the training set? (Line 425) This implies that the model does not have the generalisability that the authors claim, as it is trained on all available ROIs. How does the classification perform on ROIs where data was not included in the training set? How does a generated model perform on data that was acquired in a separate run/experiment? Answering these two questions would allow the authors to make claims about generalisability.

Reviewer #3:

Remarks to the Author:

This article entitled "AnnoSpat annotates cell types and quantifies cellular arrangements from spatial Proteomics" would like to validate AnnoSpat for annotation of data deriving from single cell proteomic assays as IMC and Co-Detection by Indexing (CODEX) for type 1 Diabetic (and non-diabetic organ- donor cohorts). They show that it is capable to recapitulate islet pathobiology and show differential dynamics of PP cells other than CD8 positive T cell abundance in the infiltrate, during diabetes progression. The developed methodology could have important implications for spatial proteomics data interpretation in several fields of biomedical research, as authors show the better performances with respect to other annotators.

However, some points need to be addressed.

Specifically:

1) The study of cell-cell interactions is of capital importance to understand disease pathogenesis, but the choice of the islets of Langerhans during Type 1 diabetes onset and development up to the beta cell loss is a rather challenging example under several aspects, either due to the dynamics of the pathology and the timing of this very changeable pathological syndrome, that is different among the islets as well as completely subverts the relationship between cells among the islets. The Authors did not pay enough attention to this great problem that affects and strongly impinges the cell-to-cell interactions.

2) When Authors state that AnnoSpat accurately identifies cell types in the complex pancreatic tissue, they do not sufficiently take into consideration the differences among pancreatic tissue intended as the exocrine gland and the much more complexes endocrine islets of Langerhans. The

differences are great and require a better or easier demonstration.

3) The Authors focused on other cell types and specifically to the PP cells, while many other cytotypes are present, including somatostatin cells, gastrin cells and others (Please read the Review "Discov. Med. 2014, 18, 141–150");

4) The Authors state that AnnoSpat elucidates CD8 positive T cell infiltration in the islet during T1D development, but they did not show that, during the development of the disease those cells, along with other inflammatory cells, including macrophages (no news in this article) are at first involved to constitute the immune infiltrate, and increase in number, then, as the beta cells die, reduce in numbers and are often detectable in the phagocytic bodies inside macrophages, until they disappear. What happens then is the reduction of the inflammatory infiltrate. This dynamic is not correctly highlighted. Therefore, the study does not show all what happens during the whole process of type 1D pathogenesis (see for this point: Metabolism 1999, 48, 477–483, Diabetologia 2010, 53, 690–698; Diabetologia 2013, 56, 2541–2543, Diabetologia 2014, 57, 841, Diabetes 2016, 65, 719–731, Diabetologia 2016, 59, 492–50;). On the contrary, authors claim to identify "potential differences in immune responses during T1D progression (lines 301-302). In other words, once islands are destroyed there are no antigens left to attack! So this explains the absence of T cells in the prolonged T1D group.

5) Another extremely important aspect, neglected in this study, is related to the very important role exerted by blood flow, islet capillaries and endothelial cells, for the so many cell to cell interactions during immune infiltration not only because the infiltrating cells are mainly recruited through peri-islet and intra-islet vessels. The blood flow changes and impinges the cell-to-cell interactions during T1D as demonstrated (see the specific studies suggested below).

Their important role must, therefore, be taken into consideration as it has been highlighted either in old and in very recent studies published also in this Journal (please see: Proc. Natl. Acad. Sci. USA 2015, 112, 1511–1516; Nat. Commun. 2018, 9, 1742; Cells 2022, 11, 3941; Cell 2020, 180, 1198–1211)

6) The Discussion must, then, take into consideration what highlighted above.

7) A minor point is related to lines 147-149 where authors state that endocrine populations have antibodies with higher quality...please better explain this sentence.

8) The article is rather complex and often the Authors write "such as" repeatedly. A more fluent writing is required.

We thank the reviewers for their thorough and constructive critique of our study. After addressing the reviewers' questions, the main conclusions of our study remain the same:

1. Several benchmarking experiments demonstrate that AnnoSpat's Annotator function can efficiently and accurately automate cell-type annotation in the absence of manually labeled training data.
2. AnnoSpat's Spatial Pattern Finder effectively quantifies known spatial relationships among cell types.
3. AnnoSpat is capable of faithfully replicating established changes in the pancreas during T1D natural history, including loss of beta cells and dynamics of aggregation of CD8+ T cells with islets.

We have fully addressed the comments and questions of the reviewers in the revised manuscript. The reviewers' critiques inspired us to perform new experiments, including new comparative analysis with K-means clustering-based cell-type annotation and HistoCAT, as well as cell-type prediction in two new ROIs from the most recent Human Pancreas Analysis Program donors. As a result of our new data, we added 45 and revised 3 figure panels in the revised manuscript. Before addressing each of the specific points, we would like to outline the notable new experiments and data included in the revised manuscript:

1. We now included cell-type annotation in two new IMC experiments from the most recent Human Pancreas Analysis Program donors that were not part of the 143 slides of 19 donors that were analyzed in the original submission. This analysis clearly shows the ability of trained AnnoSpat's model to accurately predict cell types in IMC experiments.
2. We now included several analyses examining AnnoSpat's robustness to initialization parameters. Together, these new analyses show that AnnoSpat is robust to initialization of known cell-type marker protein list, and cluster centroid calculation.
3. We now included comparative analysis with HistoCAT. This new data shows that AnnoSpat outperforms HistoCAT in faithfully replicating the established dynamics of CD8 T cell infiltration within islets during the natural history of T1D.
4. We now included comparative analysis with unsupervised K-means clustering-based cell type annotation. This data shows inferior performance of K-means clustering compared to all other supervised cell-type annotators, including AnnoSpat.

New experiments and updated text:

1. *New main figures:* 2A-C, 3C-F, 7J-7N.
2. *Revised main figures:* 2A-C (original 2A-C with new data).
3. *Reordered main figures:* 3G and 3H (original 3C and 3D), 3I and 3J (original S4C and S4D), 4A (original 3E), 4B and 4C (original 3F and 3G).
4. *New supplemental figures:* S1A-C, S1G, S2D-H, S3A-H, S4D.
5. *Reordered supplemental figures:* S1D-F (original S2A-C), S2A-C (original S1A-C), S4A and S4B (original S2D and S2E), S4C (original S5A-F), S5A-J (original S3A-J),
6. Extensive changes to the text responding to the reviewers' questions were highlighted in blue.

Below we provide a detailed point-to-point response to specific reviewers' comments.

Reviewer #1 (Remarks to the Author)

In this manuscript, Mongia et al. have developed AnnoSpat which identifies cell types and performs cell-cell proximity analysis using spatial proteomics data obtained from IMC and CODEX techniques. The authors combined semi-supervised and supervised learning methods for cell type annotation on IMC data from pancreatic tissues, and they compared AnnoSpat's performance with other methods such as SSC, SCINA, AUCell, and expert-annotated results. For the spatial analysis, the authors applied point process theory to model cell-cell relationships at different distances. And they found the spatial patterns between CD8+ T cell and islets during the progression of type 1 diabetes (T1D). However, there are some concerns that need to be addressed by the authors as follows:

We greatly appreciate the reviewer's recommendations and addressed all the specific comments in the following.

1. If we believe that semi-supervised constrained K-means clustering results are reliable and use them as the training dataset, why not employ this strategy for the entire dataset? The authors should explain the rationale behind why it is necessary to perform the cell annotation using the 2-stage approach, and how does it bring additional advantage over the single-stage approach in SSC?

This comment inspired us to further clarify the rationale behind AnnoSpat design in the revised manuscript and perform experiments to demonstrate the utility of classification (e.g. AnnoSpat) over clustering-based (e.g. K-mean or semi-supervised K-mean) cell-type annotation methods.

Clustering-based cell-type annotations partition cells based on their protein expression profiles and assign cell types to groups of cells using differential protein expression analysis. Although this approach could be effective for small studies, they fail to scale to large atlases such as the Human Pancreas Analysis Program (HPAP), where additional data is continuously collected. Upon availability of new IMC / CODEX data, current clustering-based methods must reanalyze the entire dataset for annotating the cells in the new experiment. In contrast, classifier-based cell-type annotators take advantage of their trained model to eliminate laborious reanalysis of the entire dataset and efficiently predict cell types for new IMC / CODEX samples.

To illustrate this scenario, we evaluated the ability of AnnoSpat to accurately predict cell types for two new IMC experiments from donors that were not part of the cohort at the time of original submission (see Reviewer 2's Comment 5).

SSC is a variant of AnnoSpat with its classifier replaced by centroids from the semi-supervised clustering step. Comparative analysis presented in the original submission showed additional benefits of AnnoSpat over SSC. Here, we included some of the reported observations. In the original submission, we reported that "SSC- but not AnnoSpat-annotated delta cells expressed high levels of CD14, a protein expressed in macrophages and not delta cells" (now Figures 2B and 2C rows 2 and 4; new figures S1B and S1C rows 1 and 3). We also reported that "AnnoSpat consistently outperformed SSC in predicting abundant endocrine cells from CODEX data. For instance, ghrelin, a canonical marker of epsilon cells, was highly expressed in SSC-labeled delta cells (now Figure S5C and S5D), supporting the advantage of ELM usage in AnnoSpat."

Our new analysis further confirmed the advantage of AnnoSpat over SSC. According to KL divergence, SSC failed to detect expert cell-type mislabeling in (HPAP002, Head) (Figures 3A, 3B, and 3C)

2. The authors should provide explain why using ELM in the second-stage cell type classification because it looks very random. How important is this selection? Note that speed is not the ultimate concern in terms of cell annotation tasks. In addition, there are many other classifiers available which don't perform backpropagation.

We agree with the reviewer that there are ample classification algorithms which do not require backpropagation. As described in the original submission, we selected ELM over other methods because it is not only faster but also is a neural network classifier that does not require tuning and backpropagation, and provides generalization performance.

During the design phase of AnnoSpat, we had compared ELM with SVM, decision trees and multi-layer perceptrons. Per Reviewer 1 suggestion and considering the available space in the manuscript, we only included the results of cell-type annotation using SVM in this revision. This comparative analysis shows that AnnoSpat annotated ~1,170,000 cells in diabetes IMC dataset ~2-folds faster than SVM (73.9 vs 159.9 sec) with comparable accuracy (Figure S4D). Given nearly equal performance, we used the faster algorithm for AnnoSpat implementation. In the revised manuscript, we also clarified that cell-type annotation speed is important for atlas scale applications where near online analysis can contribute to deriving insights from data as it becomes available.

3. The authors should also include unsupervised clustering strategies in their performance comparison rather than constraining the comparison between supervised methods without training data. Ultimately, the goal of cell type annotation is to identify individual cell types, and unsupervised methods are still the dominant methods in the literature for this goal. So it is important to prove the proposed method provides benefits over existing methods.

Per Reviewer 1 suggestion, we included comparative analysis with K-means clustering-based cell type labeling in the revised manuscript. As new Figures 2, S1, and S2 show, almost all supervised methods included in our comparative analysis outperformed K-means clustering-based cell-type prediction. For instance: somatostatin, PPY and CD8 were highly expressed in AnnoSpat-labeled, SSC-labeled, SCINA-labeled and AUCell-labeled, but not in K-mean-labeled delta, PP and CD8+ T cells, respectively (Figures 2, S1, and S2). Together, this data confirms studies in Geuenich et al. and further support benefits of supervised- and marker-based cell-type annotation methods over unsupervised K-means clustering approaches.

4. In the result section "AnnoSpat accurately identifies cell types in complex pancreatic tissues", there are some immune cell types in the T1D dataset but why the authors only compared the annotation results for endocrine cells (from Line 119 to 122). Importantly, it should include in the evaluation more challenging cell types, e.g., cell types that can somehow overlapped such as CD4 T vs T reg, or cell types that show only subtle differences, e.g., effective T vs memory T, etc.

We agree with the Reviewer 1 and emphasize that our choices for benchmarking was dictated by the antibodies' specificity. Indeed, we examined the ability of AnnoSpat in detecting "similar" cell types from the same lineage in the original submission (please see Lines 192-210).

We discussed the impact of antibody specificity on immune cell annotation in the original submission: in addition to endocrine cells, AnnoSpat effectively detected other cell types that had high quality antibodies and

are commonly present in the pancreatic tissue (new Figures S1D-F and S2C-H). For instance, AnnoSpat clearly identified CD8+ T cells that had a specific antibody (new Figures S1D-F and S2C-H). Conversely, detection of helper and memory T cells was less accurate due to lack of CD45RO and CD4 antibody specificity included in the HPAP IMC antibody panel by nearly all the algorithms included in our comparative analysis (new Figures S1D-F and S2C-H).

Like any other antibody-based assay, quality of IMC data is dependent on sensitivity and specificity of antibodies used in the experiments. As shown in the original Figure S2 (new Figure S1D-F), the quality of CD45RO and CD4 antibodies was far from ideal to form the basis of a reliable benchmarking analysis. As noted in the original submission, we specified that alpha, beta, PP, delta, and epsilon cells were used for benchmarking primarily because “compared to other cell types, these endocrine populations were particularly suitable for comparative analysis due to higher quality of their antibodies.” In response to Reviewer 1’s great suggestion, we highlighted the rationale for focusing on endocrine cells for benchmarking analysis in an earlier part of this section.

To further demonstrate the impact of HPAP IMC assay antibody specificity on cell-type annotation performance, we quantified protein expression heterogeneity with the SI analysis and inspected protein expression profiles of cells predicted to be cytotoxic or regulatory T cells (Figures S1G and S2C-H). SI analysis showed that AnnoSpat and SSC outperform other algorithms in identifying CD8+ T cells (Figure S1G), which had a specific antibody in the HPAP IMC panel. SI analysis also suggested that AnnoSpat and Astir equally outperformed other cell type annotators in labeling regulatory T cells (Figure S1G), however close examination of protein expression profiles revealed that Astir-annotated and AnnoSpat-annotated regulatory T cells had high levels of beta actin and pS6 expressions, respectively (neither of these two proteins should be detected in T regs) (Figure S2C and S2F). In tandem with the analyses in the original submission, these new analyses show that as expected, specificity of antibodies used in antibody-based spatial proteomic studies impacts the ability of algorithms to accurately annotate cell types, and hence should be accounted for in the design of benchmarking experiments.

5. It is not very straightforward to visually compare the performance between different methods using the heatmaps showing marker proteins for endocrine cells (e.g. Figure 2A, B, and C). It could be better if the authors can quantify the results.

Unfortunately, we could not label the x-axis with protein names due to space limitations in Figure 2. To assist in evaluating the expression of 33 proteins in 5 cell types annotated by six different algorithms, we included bar graphs showing protein expression quantification in new Figure S1 per Reviewer 1’s suggestion. We also labeled the x-axis in heatmaps in Figures S2 and S3. These representations of the data should mitigate some of the challenges in exploring and comparing this multi-variable data.

6. In the result section “AnnoSpat improves the accuracy of cell type identification in expert-annotated pancreata”, how were the expert-annotated cells obtained? Were the cells annotated by experts manually or from the original paper based on a set of Boolean rules?

As noted in the revised Materials and Methods section (Lines 667-680), Wang et al. borrowed approaches commonly used in flow cytometry data analysis to manually annotate the cells in each individual ROI. In summary, the cells were annotated by a combination of manual gating and thresholding based on Gaussian mixture models that were used to separately identify positive and negative cutoffs for each individual protein channel in each ROI, followed by so-called Boolean rules. These rules start assigning the cells to alpha, beta,

delta, PP, epsilon, followed by other cell types with available markers in HPAP IMC assay, in that order, based on "Positive" and "Exclusion" markers as listed in Wang et al. For each (donor, section), we used minimum Kullback-Leibler (KL) divergence to determine distance between the Wang et al. reported endocrine cell type distribution and AnnoSpat-, SSC-, Astir-, SCINA-, or AUCell-annotated endocrine cell type distribution.

7. The mark cross-correlation function across ROIs is a method for systematic quantification for all images and also an appropriate visualization method to compare cell-cell relationships between different groups. However, the spatial patterns from AnnoSpat are the cell-cell aggregation or dispersion, which is similar to neighborhood analysis (HistoCAT, <https://www.nature.com/articles/nmeth.4391>, R implementation: <https://github.com/BodenmillerGroup/neighborhood>,). So I am wondering how AnnoSpat gains additional meaningful cell-cell relationships that cannot be found in neighborhood analysis from HistoCAT?

This suggestion raises an excellent point in comparison and corroboration with another tool. Therefore, we included neighborhood analysis with HistoCAT and compared the results with AnnoSpat. We used HistoCAT's default parameters and the 'testInteractions' function, which returns interaction count ('ct') between two cell types. To compare with AnnoSpat, we generated analysis results similar to that of AnnoSpat, by using 'ct' instead of distance-dependent (r) mark cross-correlation function ($k(r)$).

HistoCAT and AnnoSpat reported similar spatial relationships between alpha, beta, delta, PP, epsilon and exocrine acinar cells (Figures 7A-D, 7J-M). By contrast, CD8+ T cells and islets spatial relationship patterns quantified by HistoCAT and AnnoSpat were markedly different (Figures 7N and 7E). While AnnoSpat's mark cross-correlation function median was significant different between control, AAb+ and recent T1D donors, the median of HistoCAT's interaction counts were invariant across the four disease conditions, suggesting that HistoCAT is unable to distinguish changes in aggregation of CD8+ T cells with islets during T1D progression. Notably, HistoCAT analysis failed to discern a significant difference between the levels of islets CD8+ T cell infiltration of control and recent T1D samples ($p > 0.05$). This marked difference between AnnoSpat and HistoCAT prediction could be due to their divergent approach to quantify inter-cell-type proximity. HistoCAT uses a fixed neighborhood definition of three nearest neighboring cells. By contrast, AnnoSpat considers a range of distances. Together, these spatial aggregation analysis in T1D and healthy donors demonstrates the utility of AnnoSpat in quantification of inter-cell spatial relationships.

Reviewer #2 (Remarks to the Author)

The manuscript by Mongia et al. titled “AnnoSpat annotates cell types and quantifies cellular arrangements from spatial proteomics” presents a method for annotation and quantitative localisation of cell types in intact tissues. While the authors present seemingly successful and interesting results, the methodology itself includes many opaque parameters and appears dependent on good quality prior information, suggesting that it may not generalise well to other applications outside of those presented. More detailed comments can be found below.

We are pleased that the Reviewer 2 found our results interesting. We thank the reviewer for great suggestions and performed new experiments to address all the specific comments as described below.

1- It seems to me that the success of this method is extremely dependent on the initial clustering, but there is limited results and discussion on the effect of poor (or sub-optimal) initialisation. For example, if one cell-type is omitted from the initialisation, how does this affect the classification of the remaining cell-types? Conversely, if a cell-type is included in the initialisation, that is not present in the data, how does this affect the classification of the remaining cell-types?

We thank Reviewer 2 for this excellent suggestion, which inspired us to further evaluate the impact of initialization on AnnoSpat performance. To examine how exclusion or inclusion of cell-types in the initialization step affects AnnoSpat's performance, we performed two experiments. First, we removed NK and myeloid cells from the marker protein list and predicted identities of $n = 1,170,001$ cells measured by IMC in T1D and healthy pancreata. Second, we added myeloid cells back to the AnnoSpat initialization and repeated the cell type annotation. The first analysis showed that removal of two cell types from AnnoSpat's initialization has a marginal impact on its performance in detecting cell types with specific antibodies (Figure S3A). For example, cells annotated as alpha, beta, delta, PP, epsilon, acinar, and ductal after removal of NK and myeloid cells from AnnoSpat's initialization expressed high levels of their respective marker proteins (glucagon, c-peptide, somatostatin, PPY, ghrelin, pS6, and CA2,). The second analysis showed that inclusion of a cell type in AnnoSpat's initialization also had a marginal impact on its performance (Figure S3B): myeloid-annotated cells after inclusion of myeloid markers in AnnoSpat's initialization expressed high levels of CD68.

2- It is also not clear how the user should select the quantile q_{high} for each cell type. This appears to be arbitrarily selected – can the authors comment on the effect setting this parameter? What happens if this is set too low? (what effect does excluding data/introducing a bias in the representative cells have on the classification) And too high? (what effect does including ‘outliers’ have on the classification)

To further evaluate the impact of AnnoSpat's initialization on performance, we examined how changes in cluster centroid initialization by q_{high} impacts protein expression profiles of annotated cell types (Figure S3C-G). In line with recommended default parameters (see Materials and Methods), the impact of q_{high} on performance depended on cell-type abundance and antibody specificity. This analysis showed that a more relaxed q_{high} (i.e. lower values) decreased the accuracy of cell-type prediction for both abundant and rare cell types, with a greater impact on rare cell populations (Figures S3E-G). For instance, using a more relaxed q_{high} led to markedly lower levels of c-peptide and ghrelin in cells predicted as beta and epsilon in the Combined cohort, respectively. In contrast, a more stringent q_{high} (i.e. higher values) had marginal impact on performance, especially for abundant cell types, even with moderate antibody specificity, like acinar cells (Figures S3C-E). Together, this analysis clarifies how expected cell population abundance could guide AnnoSpat initialization by showing that too relaxed and stringent q_{high} values could impact cell-type

accuracy by potentially including non-representative cells and measurement artifacts in a training dataset, respectively.

3- How is M_c calculated for the “unknown” class? If it is truly unknown, then equation on line 473 cannot be used, as M_+ and M_- is not known (and indeed could be contradictory if multiple cell-types are captured in “unknown”).

We thank the reviewer for pointing out this oversight. We now revised the Materials and Methods section (Lines 598-614) and included the procedure for identification of representative cells for the “unknown” class.

4- How was concordance measured for Figure 3A vs 3B? It is not clear how the authors concluded that there is 80% agreement between expert and AnnoSpat. How do the other methods evaluated (SSC, Astir, SCINA, AUCell) perform when compared to the expert annotations?

To address this excellent comment, we used Kullback-Leibler (KL) divergence to quantitatively compare expert-annotated with AnnoSpat-, SSC-, Astir-, SCINA-, and AUCell-annotated endocrine cell composition distributions (Figure 3A-F). This analysis revealed that AnnoSpat outperforms other algorithms in detecting samples with expert cell-type mislabeling. According to KL divergence, SSC and SCINA failed to detect expert cell-type mislabeling in (HPAP002, Head). In addition to (HPAP006, Head) that was correctly flagged, Astir-based analysis incorrectly denoted expert cell-type mislabeling in (HPAP014, Body) and (HPAP015, Tail), which was due to Astir's failure in detection of PP and epsilon cells the ROIs for these two patients. AUCell had the lowest accuracy and incorrectly flagged discordance in 14 out of 15 (93%) IMC samples included in the analysis. For further discussion please refer to Lines 291-313.

5- Why were 50% of the cells from each ROI included in the training set? (Line 425) This implies that the model does not have the generalisability that the authors claim, as it is trained on all available ROIs. How does the classification perform on ROIs where data was not included in the training set? How does a generated model perform on data that was acquired in a separate run/experiment? Answering these two questions would allow the authors to make claims about generalisability.

We thank the reviewer for pointing out this oversight. We now have revised the Materials and Methods (Lines 534-538) to clarify that AnnoSpat uses all the cells in 50% of ROI for its training set (as was shown in the Figure 1 diagram) rather than 50% of cells in all ROIs. This clarification highlights the potential application of AnnoSpat to other data sets.

To demonstrate AnnoSpat generalizability, we also provided new analysis in the revised manuscript where the trained AnnoSpat model is used to annotate cells in 2 new ROIs from the most recent Human Pancreas Analysis Program donors. This analysis clearly shows the ability of AnnoSpat to accurately predict cell types in IMC experiments that were not part of 143 slides of 19 donors that were analyzed in the original submission (Figure S3H). Inspection of protein expression profiles of AnnoSpat-annotated alpha, beta, delta, PP, epsilon, and CD8+ T cells in these two new ROIs showed high levels of glucagon, c-peptide, somatostatin, PPY, ghrelin, and CD8 expressions, respectively.

Reviewer #3 (Remarks to the Author)

This article entitled “AnnoSpat annotates cell types and quantifies cellular arrangements from spatial Proteomics” would like to validate AnnoSpat for annotation of data deriving from single cell proteomic assays as IMC and Co-Detection by Indexing (CODEX) for type 1 Diabetic (and non-diabetic organ- donor cohorts). They show that it is capable to recapitulate islet pathobiology and show differential dynamics of PP cells other than CD8 positive T cell abundance in the infiltrate, during diabetes progression. The developed methodology could have important implications for spatial proteomics data interpretation in several fields of biomedical research, as authors show the better performances with respect to other annotators.

However, some points need to be addressed.

Specifically:

1- The study of cell-cell interactions is of capital importance to understand disease pathogenesis, but the choice of the islets of Langerhans during Type 1 diabetes onset and development up to the beta cell loss is a rather challenging example under several aspects, either due to the dynamics of the pathology and the timing of this very changeable pathological syndrome, that is different among the islets as well as completely subverts the relationship between cells among the islets. The Authors did not pay enough attention to this great problem that affects and strongly impinges the cell-to-cell interactions.

We acknowledge and agree with Reviewer 3 that there exist many challenges that impede progress in comprehending the pathology of type 1 diabetes. The degree of heterogeneity within the pancreas and type 1 diabetes in effect assist in our refinement of AnnoSpat, aiding in our validation due to the distinct cell types present in these samples across non-T1D samples. From these controls, we were able to both benchmark AnnoSpat and confirm its performance for use in addressing long-standing questions of progression in T1D. By demonstrating the accuracy and applicability of AnnoSpat, we hope to power future investigations into T1D IMC data.

We are delighted to learn that Reviewer 3 found our analysis to be convincing, and noted that AnnoSpat "is capable to recapitulate islet pathobiology and show differential dynamics of PP cells other than CD8 positive T cell abundance in the infiltrate, during diabetes progression." Our demonstration of AnnoSpat's utility and precision includes its ability to faithfully replicate established changes in the pancreas during the natural history of T1D. We hope that by doing so, we can encourage the research community to harness the potential of AnnoSpat for future discoveries that extend beyond the scope of our paper, which primarily introduced this innovative software tool.

2) When Authors state that AnnoSpat accurately identifies cell types in the complex pancreatic tissue, they do not sufficiently take into consideration the differences among pancreatic tissue intended as the exocrine gland and the much more complexes endocrine islets of Langerhans. The differences are great and require a better or easier demonstration.

We thank the reviewer for this suggestion. As mentioned above, we aimed to provide analysis that can establish accuracy of AnnoSpat software. Hence, we limited our benchmarking studies to cell types whose markers had high sensitivity and specific antibodies in the HPAP IMC assay. Like any other antibody-based assay, quality of IMC data is dependent on sensitivity and specificity of antibodies used in the experiments. For example, as shown in the original Figure S2 (new Figure S1D-F), the quality of CD45RO and CD4 antibodies was far from ideal to form the basis of reliable benchmark tests. As noted in the original submission, our

choices for benchmarking were dictated by the antibodies' specificity. Indeed, we examined the ability of AnnoSpat in detecting other cell types (please see Lines 192-210).

We discussed the impact of antibody specificity on immune cell annotation in the original submission: in addition to endocrine cells, AnnoSpat effectively detected other cell types that had high quality antibodies and are commonly present in the pancreatic tissue (new Figures S1D-F and S2C-H). For instance, AnnoSpat clearly identified CD8+ T cells that had a specific antibody (new Figures S1D-F and S2C-H). Conversely, detection of helper and memory T cells was less accurate due to lack of CD45RO and CD4 antibody specificity included in the HPAP IMC antibody panel by nearly all the algorithms included in our comparative analysis (new Figures S1D-F and S2C-H).

Like any other antibody-based assay, quality of IMC data is dependent on sensitivity and specificity of antibodies used in the experiments. As shown in the original Figure S2 (new Figure S1D-F), the quality of CD45RO and CD4 antibodies was far from ideal to form the basis of a reliable benchmarking analysis. As noted in the original submission, we specified that alpha, beta, PP, delta, and epsilon cells were used for benchmarking primarily because "compared to other cell types, these endocrine populations were particularly suitable for comparative analysis due to higher quality of their antibodies." In response to Reviewer 1's great suggestion, we highlighted the rationale for focusing on endocrine cells for benchmarking analysis in an earlier part of this section.

To further demonstrate the impact of HPAP IMC assay antibody specificity on cell-type annotation performance, we quantified protein expression heterogeneity with the SI analysis and inspected protein expression profiles of cells predicted to be cytotoxic or regulatory T cells (Figures S1G and S2C-H). SI analysis showed that AnnoSpat and SSC outperform other algorithms in identifying CD8+ T cells (Figure S1G), which had a specific antibody in the HPAP IMC panel. SI analysis also suggested that AnnoSpat and Astir equally outperformed other cell type annotators in labeling regulatory T cells (Figure S1G), however close examination of protein expression profiles revealed that Astir-annotated and AnnoSpat-annotated regulatory T cells had high levels of beta actin and pS6 expressions, respectively (neither of these two proteins should be detected in T regs) (Figure S2C and S2F). In tandem with the analyses in the original submission, these new analyses show that as expected, specificity of antibodies used in antibody-based spatial proteomic studies impacts the ability of algorithms to accurately annotate cell types, and hence should be accounted for in the design of benchmarking experiments.

3) The Authors focused on other cell types and specifically to the PP cells, while many other cytotypes are present, including somatostatin cells, gastrin cells and others (Please read the Review "Discov. Med. 2014, 18, 141-150");

We appreciate the Reviewer's comment, and agree that a multitude of cell types contribute to T1D pathology and pathogenesis. In fact, many members of our multidisciplinary team contributed the results summarized in the review by Dr. Pugliese (PMID: 25227755). As noted earlier, the aim of this manuscript is to present a novel software tool that can be leveraged for analysis of spatial proteomics data in future T1D studies. We particularly focused on T1D due to the complexity of cell types involved in the disease pathobiology as noted by Reviewer 3. As noted in response to Reviewer 3's Comment 2, we specifically focused on analyses that can demonstrate AnnoSpat's accuracy while utilizing the unique features of the HPAP IMC datasets. To this end, all our analyses aimed to evaluate whether AnnoSpat results could recapitulate known T1D phenotypes. Antibodies included in the IMC panel limited our ability to examine many cell types discussed in the review by Dr. Pugliese (PMID: 25227755). For instance, HPAP IMC and CODEX datasets do not allow examination of

epitope spreading and differential number of islet-reactive CD8 T cell populations in islets. Quality and specificity of antibodies used in the HPAP IMC and CODEX panels further confounded our analysis that can be used for accurate benchmarking analysis as noted above.

4) The Authors state that AnnoSpat elucidates CD8 positive T cell infiltration in the islet during T1D development, but they did not show that, during the development of the disease those cells, along with other inflammatory cells, including macrophages (no news in this article) are at first involved to constitute the immune infiltrate, and increase in number, then, as the beta cells die, reduce in numbers and are often detectable in the phagocytic bodies inside macrophages, until they disappear. What happens then is the reduction of the inflammatory infiltrate. This dynamic is not correctly highlighted. Therefore, the study does not show all what happens during the whole process of type 1D pathogenesis (see for this point: Metabolism 1999, 48, 477–483, Diabetologia 2010, 53, 690–698; Diabetologia 2013, 56, 2541–2543, Diabetologia 2014, 57, 841, Diabetes 2016, 65, 719–731, Diabetologia 2016, 59, 492–50;). On the contrary, authors claim to identify “potential differences in immune responses during T1D progression (lines 301-302). In other words, once islands are destroyed there are no antigens left to attack! So this explains the absence of T cells in the prolonged T1D group.

As Reviewer 2 summarized, the immune response in T1D autoimmunity is indeed complex. To clarify our results, we extensively reworded this section and referred to the suggested publications. We have also updated the conclusion and title of our benchmarking analysis that shows AnnoSpat is capable of recapitulating the expected dynamics of CD8+ T cells aggregation with islets during the natural history of T1D. Unfortunately, the HPAP IMC assay does not have the resolution and necessary antibodies to differentiate various monocytes and phagocytic bodies inside macrophages. In fact, our new comparative analysis clearly shows that HistoCAT (another method designed for capturing cell-cell proximity) fails to recapitulate the this dynamic.

Below is a summary of publications recommended by Reviewer 3. Per Reviewer 3's Comment 6, the revised Discussion now includes a summary of these publications in relation to the description and benchmarking of AnnoSpat software.

Metabolism 1999, 48, 477–483, <https://pubmed.ncbi.nlm.nih.gov/10206441/> :

By studying the OLETF rat model, a spontaneous model of non-insulin-dependent diabetes mellitus, Mizuno et al. showed that dimorphism in OLETF diabetic phenotype is correlated with differences in islet beta cell number and islet capillaries morphology. More specifically, the authors examined microvascular morphological differences between lean and obese diabetic OLETF mice and showed that the islet capillaries and angioarchitecture from lean OLETF rats differ dramatically from their normal and obese diabetic counterparts. However, they could not rule out the possibility that the alterations in the angioarchitecture of islets observed in lean OLETF rats may be secondary to persistent hyperglycemia.

Diabetologia 2010, 53, 690–698; <https://pubmed.ncbi.nlm.nih.gov/20062967/> :

Earlier studies of childhood-onset diabetic patients with diabetes duration of greater than 5 years reported heterogeneity of insulin-positive cells in pancreatic islets. Gianani et al. characterized pancreata of type 1 diabetic children to identify the pathological basis of this heterogeneity. They reported insulin-positive cells within at least some pancreatic islets in 30% of their type 1 diabetic autopsy cohort. They further showed that heterogeneity of C-peptide secretion is explained by two distinct histological patterns: 1) Pattern A had lobular

retention of areas with surviving 'abnormal' beta cells producing survivin, an apoptosis inhibitor, and HLA class I. 2) Pattern B had 100% 'normal'-appearing islets with reduced beta cell counts without survivin or HLA class I. Interestingly, they observed differential insulinitis in pattern A and B islets. They did not identify any insulinitis in any of the autopsies with pattern B histology, while insulinitis and beta cells could be observed in 2 of 20 donors with class A histology.

Diabetologia 2013, 56, 2541–2543, <https://pubmed.ncbi.nlm.nih.gov/24006089/> :

Campbell-Thompson et al. reports criteria defined at the fifth annual nPOD meeting for diagnosis of insulinitis in T1D. Although CD8+ T cells constitute the majority of infiltrating cells in recent-onset T1D, B cells can be observed in more advanced stages of insulinitis. Hence, the count of CD45+ leukocyte cells was recommended for diagnosis of insulinitis. Interestingly, this report noted that insulinitis could be observed in (pseudo)atrophic islets devoid of beta cells. Most published literature, which are predominantly based on autopsy material, stress that insulinitis is almost always found in islets in which residual beta cells are present. More recent work utilizing materials from type 1 diabetic organ donors indicate that insulinitis lesion may be more frequent in (pseudo)atrophic islets devoid of beta cells than previously recognized.

Diabetes 2016, 65, 719–731, <https://pubmed.ncbi.nlm.nih.gov/26581594/> :

Using immunostaining in serial sections of pancreata from ~100 organ donors, Campbell-Thompson et. al. showed that although insulinitis frequency (the percentage of islets displaying insulinitis) inversely correlates with diabetes duration, the numbers of T and B cells appeared independent of islet insulin immunopositivity and diabetes duration. This study further showed that numbers of T and B cells correlates with the number of insulinitic islets irrespective of islet beta-cells. In line with earlier studies, they reported that the proportions of insulin-positive islets with insulinitis was higher than in insulin-negative islets in T1D donors, yet in contrast to earlier work they showed that insulin-negative islets were also affected by insulinitis. Together, this and other studies support the presence of beta-cells as well as insulinitis several years after diagnosis in children and young adults, and suggest that the chronicity of islet autoimmunity extends well into the postdiagnosis period. Complexity of pancreatic tissue and heterogeneity of T1D pathology underscore the need for algorithms that can facilitate high throughput and high resolution in situ examination of pancreata from type 1 diabetic organ donors. Here, we showed the ability of AnnoSpat in faithfully replicating dynamics of CD8+ T-cell aggregation with islets during the natural history of T1D. This example shows the potential utility of AnnoSpat in facilitating future characterization of T1D heterogeneity and complexity.

Diabetologia 2014, 57, 841, <https://pubmed.ncbi.nlm.nih.gov/24429579/> :

This manuscript introduced the Diabetes Virus Detection study (DiViD), which is the first study that aims to collect samples of pancreatic tissue by pancreatic tail resection from live adult patients newly diagnosed with type 1 diabetes.

Diabetologia 2016, 59, 492–50 <https://pubmed.ncbi.nlm.nih.gov/26602422/> :

By studying six pancreatic tissues from living onset diabetic patients, Krogvold et al. reported 36% insulinitis frequency. In contrast to Campbell-Thompson et. al.'s study, they observed that islet-associated T cells are located in the endocrine-exocrine interface rather than within the islet parenchyma. They concluded that the incapacity of the CD8+ T cells to invade the islet and mediate efficient beta cell lysis could potentially be explained by the extraordinarily weak affinity between the T cell receptor (TCR) and the HLA class I. The study

could guide future IMC assay design potential inclusion of tetramer staining to detect T cells with reactivity to distinct beta cell auto-antigens.

5) Another extremely important aspect, neglected in this study, is related to the very important role exerted by blood flow, islet capillaries and endothelial cells, for the so many cell to cell interactions during immune infiltration not only because the infiltrating cells are mainly recruited through peri-islet and intra-islet vessels. The blood flow changes and impinges the cell-to-cell interactions during T1D as demonstrated (see the specific studies suggested below). Their important role must, therefore, be taken into consideration as it has been highlighted either in old and in very recent studies published also in this Journal (please see: Proc. Natl. Acad. Sci. USA 2015, 112, 1511–1516; Nat. Commun. 2018, 9, 1742; Cells 2022, 11, 3941; Cell 2020, 180, 1198–1211)

Per Reviewer 3's suggestion, we discussed the importance of studying changes in capillary vessels, microvasculature morphology, and blood flow that impinged in cell-to-cell interaction during T1D pathogenesis. Unfortunately, IMC assay is not designed to measure changes in capillary vessels, microvasculature morphology, or blood flow. As noted in the revised Discussion, here, we demonstrated AnnoSpat's ability to faithfully replicate the expected dynamics of CD8+ T cell aggregation with islets during the natural history of T1D. This example demonstrates the potential utility of AnnoSpat in facilitating future characterizations of T1D heterogeneity and complexity. We postulate that AnnoSpat and other algorithmic advances will pave the way for the integration of spatial proteomics data with readouts from histopathological and microscopic assays. This integration can expand our holistic understanding of T1D pathobiology and facilitate the discovery of links between alterations in islet microvasculature, capillary morphology, endothelium, and immune infiltration with islet dysfunction.

Below is a summary of publications recommended by Reviewer 3. As per Reviewer 3's Comment 6, the revised Discussion now includes a summary of these publications in relation to the description and benchmarking of AnnoSpat.

Proc. Natl. Acad. Sci. USA 2015, 112, 1511–1516 <https://pubmed.ncbi.nlm.nih.gov/25605891/> :

Magnuson et al. studied the pancreatic infiltrate during disease progression in nonobese diabetic (NOD) mice, using a noninvasive labeling and tracking approach. As noted by the authors and others, dynamics and composition of leukocyte infiltration is significantly different in NOD mice and human patients. The proportion of infiltrated islets and the extent of infiltration appear generally lower in human patients than in NOD mice, and a dominance of CD8+ over CD4+ T cells seems frequent, with a variable frequency of B lymphocytes. Notably, this study provides data suggesting that B and T lymphocytes as well as CD11b monocytes infiltrate insulitic islets in NOD mice, and infiltrating lymphocytes are continuously replenished which was for the most part independent of pancreatic antigen-specific. In line with our observation, the authors reported marked proliferation of T lymphocytes in the T1D pancreas. By tracing trafficking of leukocytes from cervical lymph nodes to pancreas, the authors concluded that in 24 h lymphoid but not myeloid cells travel from cervical lymph nodes to the pancreas. CD4+ and CD8+ cells migrated to the pancreas at the same rate. The majority of recent CD4+ immigrants were naive antigen-inexperienced cells and not effector / memory phenotype. They also observed that T regs have limited migration to the pancreata of NOD mice. Surprisingly, they showed that migration into the autoimmune pancreas correlated in a nonspecific manner to the total insulitis size, rather than to a particular drive from antigen-specific receptors.

Nat. Commun. 2018, 9, 1742; <https://pubmed.ncbi.nlm.nih.gov/29717116/> :

Using contrast-enhanced ultrasound measurements of pancreas blood flow dynamics, the authors showed that islet blood-flow dynamics alters prior to diabetes onset in STZ-treated mice, NOD mice, and adoptive-transfer mice. However, they did not report any application of their interesting approach in human.

Cells 2022, 11, 3941; <https://pubmed.ncbi.nlm.nih.gov/36497199/> :

This excellent review summarizes the literature describing how the infiltrating immune cells reach the islets through blood, lymphatic vessels, and extra-cellular matrix spaces, and are influenced by islet micro endothelium. It also highlights that relation between insulinitis and microvasculature reorganization and blood flow changes during type1 progression is not fully understood. Hence, future IMC analysis with antibodies for the markers of these cell types in conjunction with histopathological and microscopy assays could be used to further examine potential links between alterations in islet microvasculature, capillaries morphology and endothelium with islet dysfunction.

Cell 2020, 180, 1198–1211 <https://pubmed.ncbi.nlm.nih.gov/32200801/> :

This interesting paper describes protein C receptor positive (Procr+) cell population in adult mouse pancreas, which are multipotent endocrine progenitor cells and can generate all for major endocrine cell types. The Procr antibody does not exist in the HPAP IMC or CODEX panels.

6) The Discussion must, then, take into consideration what highlighted above.

As noted in response to Reviewer 3's comments 4 and 5, we reworded several parts of the manuscript, referred to the suggested publication, revised the Discussion with a brief summary of key points relevant to IMC data analysis, and clarified the conclusions of our benchmarking analysis, which was aimed to demonstrates examples of AnnoSpat's capability in faithfully replicating established changes in the pancreas during the T1D natural history: loss of beta cells and dynamics of aggregation of CD8+ T cells with islets. Like all assays, IMC has its own pros and cons. For instance, HPAP IMC assay does not have the resolution and antibodies required to differentiate various monocytes and phagocytic bodies inside macrophages.

We also revised the manuscript to discuss the importance of alterations in capillary vessels, microvasculature morphology, and blood flow that impinged on cell-to-cell interaction during T1D pathogenesis. Unfortunately, the IMC assay is not designed to measure these phenotypes, hence we are not able to test AnnoSpat's ability to recognize these changes from HPAP IMC data. As noted in the Discussion, we postulate that AnnoSpat and other algorithmic advances will pave the way for the integration of spatial proteomics data with readouts from histopathological and microscopic assays. This integration can expand our holistic understanding of T1D pathobiology and facilitate the discovery of links between alterations in islet microvasculature, capillary morphology, endothelium, and immune infiltration with islet dysfunction.

7) A minor point is related to lines 147-149 where authors state that endocrine populations have antibodies with higher quality...please better explain this sentence.

Per Reviewer 3's suggestion, we revised the manuscript as follows:

Like any other antibody-based assay, IMC data quality highly depends on sensitivity and specificity of antibodies used in the assay. Compared to other cell types, alpha, beta, PP, delta, and epsilon cells were

particularly suitable for comparative benchmarking analysis due to higher specificity of their antibodies in the HPAP IMC assay

8) The article is rather complex and often the Authors write “such as” repeatedly. A more fluent writing is required.

We revised the manuscript to reduce the repetition of the phrase “such as”.

Reviewers' Comments:

Reviewer #1:

Remarks to the Author:

The authors have made substantial improvements on the manuscript and have addressed many review comments with additional experiments and results. However, although most of the questions raised in the previous review have been addressed, the reviewer feels that several important questions remain unresolved in the current manuscript.

1) The evidence that AnnoSpat is better than unsupervised based methods such as K-means is not convincing enough based only on visual inspection on certain cell types. For example, if we look at the Silhouette index AnnoSpat is not clearly better compared with AUCcell or K-means?

2) In terms of algorithm design, it is not clear how AnnoSpat achieves better performance compared to K-means? Not that different from other methods such as Astir, AnnoSpat's prediction model is trained on results from K-means on the training set. In this case, how AnnoSpat achieve a different and better performance compared to K-means?

3) The authors should still extend the comparison to current single-cell annotation methods based on unsupervised clustering such as Seurat, PhenoGraph, FlowSOM, etc rather than only K-means clustering in the current manuscript. Particularly, K-mean based clustering is not commonly used in single-cell analysis as it is normally not able to capture the fine structure of the distribution of single-cell in the high-dimensional space.

Regarding previous points raised by reviewer#2 (unavailable to review in this review round):

1. Question 1: It is very useful that authors include new the experiments to analyze the effects of initialization on the performance of AnnoSpat. However, it is not sufficient to analyze the results based solely on marker gene expression of the annotation results, which doesn't show, quantitatively, difference in cell annotation results. Instead, it would be more useful to compare the annotation results using tools such as Sankey diagram so that the effects of initialization on AnnoSpat can be better illustrated.

2. Question 2: Like previous Question, it is not sufficient to visualize the results based on marker protein expression only. Tools like Sankey diagram would better reveal how these parameters affect the cell annotation results.

3. Question 4: 1) The KL results were not shown in Fig. 3 in the revised manuscript? 2) The comparison between expert and algorithm annotation is very interesting but the results are in fact a bit inconsistent. On one hand, the authors compared the performance of different annotation results based on their KL divergence to the expert annotation while on the other hand they questioned the expert annotation results when the KL divergence showed discrepancies between AnnoSpat and expert annotations? The only conclusion that can be draw from the current the result should be "although AnnoSpat show better consistency with expert annotation, close examination shows that mislabeling can happens in expert annotations"?

Reviewer #3:

Remarks to the Author:

The Authors in their revised version of the paper have addressed all the previous comments made by myself as well as by the other reviewers and they decided to perform new experiments as well as to amend the whole manuscript considerably improving it.

Now data are significantly more robust and the paper is stronger and deeper than the previous version.

The paper is very accurate either in the performed analyses than in the new added paragraphs, taking into account that they have included cell-type annotation in two new ICM experiments from the most recent Human pancreas Analysis Program

donors.

It is also very important that they have included comparative analysis with HistoCAT. All the new Figures and the whole revised manuscript provide higher significance to the study.

Specifically, regarding the role of microvessels and their alterations during T1D, the Authors have revised the manuscript to discuss this issue and the fact that blood flow impacts islet cells and their survival. These issues will be solved in a near future being now a limitation of the technique.

This referee acknowledges all the amendments and this new version of the manuscript.

Only some letters were lost in the written manuscript (spelling).

We are thrilled to learn that Reviewer 3 concluded that our response to all the reviewers was satisfactory. We also appreciate Reviewer 1 for raising additional questions about our study, which allowed us to clarify that only AnnoSpat simultaneously provides accuracy, speed, and completeness in annotating single cells measured by IMC. After addressing Reviewer 1's questions, the main conclusions of our study remain the same.

1. Several benchmarking experiments collectively demonstrate that AnnoSpat's Annotator function can rapidly and accurately automate the annotation of a large fraction of cells in IMC data, even in the absence of manually labeled training data.
2. AnnoSpat's Spatial Pattern Finder effectively quantifies known spatial relationships among cell types, providing additional insights from spatial proteomic data.
3. AnnoSpat detects established changes in the pancreas during T1D natural history, including loss of beta cells and dynamics of aggregation of CD8+ T cells with islets.

In response to Reviewer 1's latest comments, we performed new comparative analysis with Seurat, PhenoGraph, and FlowSOM. We also revised the text to clarify the questions raised by Reviewer 1.

Before addressing each of the specific points, we would like to emphasize that all the analyses in Figure 2 compare algorithmically predicted cell labels and not true cell labels, as true labels do not exist, and for all practical purposes, cannot be obtained for this or any other similar studies. In the absence of this information, which is a common case for many biological contexts, there is no feasible analysis that could discern the correctness of predicted cell labels aside from the examination of all 33 HPAP IMC-measured proteins.

To address the lack-of-ground-truth conundrum, which complicates many studies concerning cell-type quantification, we (Schwartz et al., Nat Methods, 2020, PMID: 32123397) and others have used cell sorting to create controlled cell admixtures with known identities (i.e., true cell type labels). With these ground truth datasets, we previously evaluated the performance of cell-type annotation algorithms for scRNA-seq. Unfortunately, this approach is not technically possible for IMC assays because we cannot sort cells unless we dissociate the tissue (defeating the purpose of studying cells in situ). Furthermore, the same section is not available after the experiment due to laser ablation of the tissue during the procedure.

We stress that no current algorithm or method can address all the challenges associated with cell-type annotation in single-cell proteomic analysis. In this manuscript, we propose a classification-based algorithm to begin addressing the unmet need for automated cell-type annotation in atlas-scale projects, such as the Human Pancreas Analysis Program (HPAP), where additional data is continuously deposited in data portals and must be analyzed with online methods.

K-means, PhenoGraph, Seurat, and other cluster methods only partition cells and do not label them. With a clustering outcome, an investigator would depend on a series of differential protein expression analyses followed by subjective interpretation to eventually assign a cell type label to all the cells within a cluster in a manual and laborious manner.

As shown in Figure S7 (formerly Figure S3H) in Revision 1, AnnoSpat's trained model can accurately predict cell types for new IMC experiments without the need for the reanalysis of the entire dataset, a capability not possible with clustering methods (i.e., Silhouette index = 0).

Below we provide a detailed point-to-point response to specific reviewers' comments.

Reviewer #1 (Remarks to the Author)

The authors have made substantial improvements on the manuscript and have addressed many review comments with additional experiments and results. However, although most of the questions raised in the previous review have been addressed, the reviewer feels that several important questions remain unresolved in the current manuscript.

We are pleased to learn that Reviewer 1 noted that we have addressed many of his/her comments with our additional experiments in Revision 1.

We greatly appreciate the reviewer's questions and addressed all the specific comments in the following.

1) The evidence that AnnoSpat is better than unsupervised based methods such as K-means is not convincing enough based only on visual inspection on certain cell types. For example, if we look at the Silhouette index AnnoSpat is not clearly better compared with AUCcell or K-means?

We believe this comment raises two distinct points: 1) Comparison between AnnoSpat and K-means, 2) Comparison between AnnoSpat and AUCCell. We separately responded to these points.

AnnoSpat vs K-means:

We appreciate Reviewer 1 for raising a number of questions, as they provided us with an opportunity to describe a major advantage of AnnoSpat over existing clustering algorithms. As explained in detail below and in the revised manuscript, while clustering strategies such as K-means may, in some cases, exhibit slightly more favorable Silhouette Index, which quantifies the extent of protein expression homogeneity in cells with a given label but not the correctness of their labels, these algorithms might fail to segregate cells based on their true lineage. In other words, a clustering algorithm could identify homogeneous cell groups; however, this grouping may not necessarily be driven by biology but rather potentially by stochastic initialization and/or IMC assay resolution, noise, and artifacts (please also refer to Comment 3 and Materials and Methods).

We have now clarified that, unlike AnnoSpat, K-means and other clustering algorithms fail in the biological re-evaluation of cells grouped in each cluster. Addressing Reviewer 1's concern has allowed us to clarify that AnnoSpat possesses a significant advantage over unsupervised clustering, partly due to its initialization procedure, which is both cell-type aware and robust to artifacts.

In the absence of ground truth (i.e., true cell-type labels), we devised a multi-faceted approach, as presented in Figures 2, S3, and other related supplementary figures, to assess algorithms' performance and usability for atlas-scale cell-type annotation:

- I. We emphasize that the Silhouette Index (SI) is a performance metric mostly used for clustering and not classification algorithms. This is because the SI only captures protein expression homogeneity within groups and, on its own, is not capable of capturing the correctness of inferred labels (i.e., correctness of cell type label prediction).

Hence, we used the SI solely to examine the extent of protein expression homogeneity in cells assigned a given label by each algorithm and not the correctness of the assigned labels. We acknowledge the reviewer's concern that slightly higher SI values could be misinterpreted as more accurate cell type predictions, even when the assigned cell type is clearly incorrect. In response to this concern, we point out many cases of such incorrect labeling in the manuscript and highlight a few of them below. Additionally, we have extensively revised the manuscript for clarification, addressing the reviewer's comment.

If the SI had been adequate for our comparative analysis, we would have reported it for the entire cell type labeling for each algorithm instead of separately for each individual population. As the table below shows, AnnoSpat has the highest SI values across the entire T1D, Control, and Combined cohorts, except for AUCCell in the T1D cohort. However, as shown in Figures 2D, S8A, and S8B; and discussed below, AUCCell fails to annotate up to 75% of cells in the T1D cohort, again indicating that SI alone is an inadequate measure of performance for classification-based cell-type annotation.

	Astir	SSC	AnnoSpat	SCINA	AUCell	Kmeans	Seurat	PhenoGraph
T1D cohort	0.268	0.401	0.402	0.309	0.407	0.400	0.359	0.315
Control cohort	0.251	0.367	0.381	0.227	0.363	0.374	0.284	0.377
Combined cohort	0.242	0.367	0.370	0.257	0.380	0.345	0.301	0.324

- II. Although the SI is easy to calculate and present, it cannot examine the correctness of labels because it is not designed to do so. In the absence of ground truth labels, the only way to evaluate the accuracy of predicted cell types was to rely on biological knowledge and examine the expression of all 33 HPAP IMC-measured proteins to assess the correctness of predicted cell types.

IMC/CODEX assays can measure 30-40 proteins. Each project designs a customized antibody panel for its IMC assay with the goal of identifying a priori known set of cell types in a tissue of interest. For example, the panel of 33 proteins included in the HPAP IMC was selected to ideally identify 16 cell types within pancreatic tissue. We examined and presented the expression of all 33 proteins in 16 predicted cell types (see Figures 2 and S1 to S7), with a primary focus on endocrine and CD8+ T cells for comparative analysis due to their antibody specificity (please refer to the response to Reviewer 1 – Comment 4 in Revision 1).

- III. We also examined the percentage of cells labeled by each algorithm. As explained in both the original submission and Revision 1, although it produced a higher SI in the T1D cohort, AUCell failed to annotate most cells included in the benchmarking analysis (see AnnoSpat vs AUCell below).

In direct response to Reviewer 1, we clarified the extent of the difference between the mean SI of AnnoSpat and K-means by summarizing Tables S4, S5, and S6, which are used to generate Figure 2. Below, we present the numeric differences between the mean SI of these two methods for the T1D, Control, and Combined cohorts:

AnnoSpat - Kmeans	T1D	Control	Combined
Alpha	-0.0652	-0.0761	-0.0991
Beta	0.0925	0.0238	0.1569
Delta	-0.0163	0.1583	-0.0412
PP	0.0617	0.0895	0.1508
Epsilon	-0.0647	-0.1608	-0.0447

In 11 out of 15 comparisons, there was no marked difference ($\text{abs}(\text{differential SI}) < 0.1$) between the mean SI values of AnnoSpat and K-means.

In 3 out of 15 comparisons, the mean SI of AnnoSpat was higher ($\text{differential SI} > 0.1$) than that of K-means (highlighted in green).

Only in 1 out of 15 comparisons, the mean SI of K-means was markedly higher ($\text{differential SI} < -0.1$) than that of AnnoSpat (highlighted in yellow): K-means-labeled Epsilon cells in the Control cohort.

In the absence of true labels, we go back to biology and check the actual protein expression of Epsilon-labeled cells by these two algorithms. Looking at the AnnoSpat-labeled Epsilon cell in the Control cohort shows that they clearly express ghrelin (the canonical marker of epsilon cells). In contrast, CD11b (a canonical marker of myeloid cells), not ghrelin, was the only distinctly expressed protein in the K-means-labeled epsilon cells, suggesting incorrect cell-type annotation by K-means despite a higher SI, suggesting incorrect cell-type annotation by K-means despite higher SI.

In the absence of ground truth, one can only conclude the following: endocrine biology tells us that this group of cells, clustered together by the K-means algorithm and labeled as Epsilon through a series of differential protein expression analyses, are indeed not epsilon cells. This conclusion is drawn because this group of K-means-clustered cells does not distinctly express the only protein (i.e., ghrelin) universally accepted as the marker of epsilon cells.

We refer Reviewer 1 to additional examples provided in the manuscript that show why going back to biology, although cumbersome, was necessary to examine the correctness of assigned labels by various algorithms.

AnnoSpat vs AUCell:

Regarding the Reviewer's question about AUCell, we emphasize that AUCell failed to annotate most cells included in the benchmarking analysis (see Figures 2D, S8A (original S4A), and S8B (original S4A)). Here, we reiterate some parts of the Revision 1 rebuttal / manuscript describing why we decided to look beyond a simple SI analysis to compare the performance of cell type annotation algorithms in the absence of ground truth.

- Based on SI analysis, AUCell exhibited comparable performance to AnnoSpat (2A-C rows 2 and 3; S1A-C, rows 1 and 2), but it failed to annotate most cells included in the benchmarking analysis (Figure 2D), potentially leading to information loss. Close examination of data revealed that AUCell more accurately labeled cell types with a larger number of marker proteins such as ductal cells (Figures S8A, S8B; Table S3), which is in line with AUCell's main focus on scRNA-seq which measures thousands of transcripts but not spatial proteomics technologies which measure tens of proteins.
- SI metric suggested that AnnoSpat and AUCell were comparable in identifying homogenous cell groups from CODEX data (Figure S9B, Table S11). Yet, a close examination of labeled cells (which is necessary because we do not know true cell labels, hence a feasible solution is to use biological knowledge to evaluate algorithmic predictions) revealed that while AnnoSpat identified a pure delta cell population, AUCell-annotated delta cells expressed high levels of CD206, ARG1, and CD4, canonical markers of macrophages and helper T cells, respectively (Figures S9C and S9G).
- AUCell had the lowest accuracy and incorrectly flagged discordance in 14 out of 15 (93%) IMC samples included in our comparison with manual labeling (Figure 3) (because AUCell fails to label most of the cells.)

2) In terms of algorithm design, it is not clear how AnnoSpat achieves better performance compared to K-means? Not that different from other methods such as Astir, AnnoSpat's prediction model is trained on results from K-means on the training set. In this case, how AnnoSpat achieve a different and better performance compared to K-means?

K-means usage within AnnoSpat and K-means-based cell type annotation are distinct.

As described in the "Generation of Initial Cluster Centroids" section, AnnoSpat implements constrained K-means and initializes cluster centroids by identifying representatives of each cell type based on the Marker Protein file. AnnoSpat cluster initialization is also capable of excluding assay artifacts from the initial cluster centroid calculation.

In contrast, K-means is randomly initialized and does not consider any information about expected cell types during cluster identification. Moreover, clustering-based cell-type labeling approaches (including K-means) do not directly predict cell types for each individual cell; instead, they assign a label to all the cells within a cluster based on an ad hoc and subjective series of differential expression analyses that can markedly impact cell type inference.

The advantage of cell-type aware initialization is evident from the results of our proposed SSC approach, a variant of AnnoSpat with its ELM classifier replaced by centroids from semi-supervised clustering.

3) The authors should still extend the comparison to current single-cell annotation methods based on unsupervised clustering such as Seurat, PhenoGraph, FlowSOM, etc rather than only K-means clustering in the current manuscript. Particularly, K-mean based clustering is not commonly used in single-cell analysis as it is normally not able to capture the fine structure of the distribution of single-cell in the high-dimensional space.

Per Reviewer 1's suggestion, we included comparative analysis with Seurat, PhenoGraph, and FlowSOM in Revision 2.

We included K-means in Revision 1's comparative analysis because it identifies a predetermined number of clusters. We emphasize that IMC/CODEX data points exist in a 30-40 protein space (e.g. 33 proteins in our HPAP IMC dataset), which is different from the 5000-6000 median genes in single-cell transcriptomic space intended for some of the methods noted by Reviewer 1. As listed in Tables S1, the HPAP IMC assay measures 33 proteins to ideally identify 16 cell types that are expected to be present in human pancreata. For most cell types, only one protein is included in the HPAP IMC assay; hence, the assay is not intended for the discovery of new cell types.

As discussed earlier, the SI cannot examine the correctness of labels. Nevertheless, the SI analysis showed that Seurat, one of the most popular tools for scRNA-seq analysis, had the worst performance among the clustering-based cell type annotators. This further underscores the differences between IMC and scRNA-seq data.

AnnoSpat vs Seurat:

To clarify the difference between AnnoSpat and Seurat, we summarized Tables S4, S5, and S6 and presented the numeric differences in mean Silhouette Index (SI) between these methods for the T1D, Control, and Combined cohorts:

Anno-Seurat	T1D	Control	Combined
Alpha	0.0025	-0.0675	-0.0869
Beta	0.0361	0.3343	0.1562
Delta	0.1712	0.1338	0.0137
PP	0.0857	0.0734	0.2789
Epsilon	-0.0805	0.013	-0.019

In 10 out of 15 comparisons, there was no marked difference ($\text{abs}(\text{differential SI}) < 0.1$) between means SI of AnnoSpat and Seurat.

In 5 out of 15 comparisons, the mean SI of AnnoSpat was higher ($\text{differential SI} > 0.1$) than that of Seurat (highlighted in green).

In 0 out of 15 comparisons, the mean SI of AnnoSpat was higher ($\text{differential SI} < -0.1$) than that of AnnoSpat (highlighted in yellow).

AnnoSpat vs PhenoGraph:

We also compared PhenoGraph with AnnoSpat. While both PhenoGraph and Seurat are based on Louvain clustering, they showed very different behavior in clustering IMC data, potentially due to differences in normalization and/or Louvain algorithm implementation. Notably, PhenoGraph, which is designed for CyTOF data, resulted in higher SI values compared to Seurat. Nonetheless, our SI analysis showed that, similar to Seurat, PhenoGraph is not superior to AnnoSpat in detecting homogeneous cell clusters.

Once again, we summarized Tables S4, S5, and S6 and included the numeric differences between the mean SI of AnnoSpat and PhenoGraph for the T1D, Control, and Combined cohorts for clarification:

Anno-PhenoGraph	T1D	Control	Combined
Alpha	-0.0431	-0.0588	0.0545
Beta	0.3481	0.0687	0.1405
Delta	0.1872	0.1151	0.1142
PP	0.0365	-0.0333	-0.025
Epsilon	-0.0974	-0.0704	-0.0555

Similar to the comparison with Seurat, there was no marked difference ($\text{abs}(\text{differential SI}) < 0.1$) between the mean SI of AnnoSpat and PhenoGraph in 10 out of 15 comparisons.

In 5 out of 15 comparisons, the mean SI of AnnoSpat was higher ($\text{differential SI} > 0.1$) than that of PhenoGraph (highlighted in green).

In 0 out of 15 comparisons, the mean SI of PhenoGraph was higher ($\text{differential SI} < -0.1$) than that of AnnoSpat (highlighted in yellow).

Only epsilon cells in T1D and Control cohorts had mean SI values close to our cutoff. In the absence of true cell-type labels, we again looked at the protein expression profiles of these cells to discern the accuracy of cell-type labeling. While AnnoSpat-labeled epsilon cells clearly expressed ghrelin, the canonical marker of epsilon cells, CD11b (a canonical marker of myeloid cells) was the only distinctly expressed protein in the PhenoGraph-labeled epsilon cells (Figures 2A, 2B, S3A, and S3B row 1). However, the cells grouped and labeled as epsilon by PhenoGraph did not distinctly express ghrelin, the canonical marker of epsilon cells, suggesting that these cells are not true epsilon cells.

Together, this analysis further supports the inability of K-means and Louvain-based clustering methods to correctly segregate rare populations in IMC data, an observation that was reported for scRNA-seq and scATAC-seq data (Schwartz et al., Nat Methods, 2020, PMID: 32123397; Schwartz et al., Cell Reports, 2021, PMID: 34433064). AnnoSpat attempts to overcome this issue by implementing a procedure that initializes cluster centroids based on representatives of each cell type instead of random centroid selection or heuristic graph partitioning.

AnnoSpat vs FlowSOM:

Lastly, our clustering of IMC data with FlowSOM showed that this algorithm failed to effectively separate the cells in the Combined, Control, and T1D cohorts, leaving 1,151,979, 793,571, and 363,961 cells in a single cluster, respectively. For instance, FlowSOM labeled only 84, 259, and 2,109 cells as Alpha cell in the Combined, Control, and T1D cohorts, respectively, which is not biologically plausible (alpha cells are among the most abundant endocrine cells in pancreas). Additionally, we were unable to perform our SI analysis for FlowSOM because it created very small clusters for many endocrine cell types, which prohibited us from calculating SI distributions needed for Figure S3A-C.

Regarding previous points raised by reviewer #2 (unavailable to review in this review round):

1) Question 1: It is very useful that authors include new the experiments to analyze the effects of initialization on the performance of AnnoSpat. However, it is not sufficient to analyze the results based solely on marker gene expression of the annotation results, which doesn't show, quantitatively, difference in cell annotation results. Instead, it would be more useful to compare the annotation results using tools such as Sankey diagram so that the effects of initialization on AnnoSpat can be better illustrated.

Per Reviewer 1's request, we included Sankey plots for presenting of data from our Marker Protein file exclusion and inclusion experiments. As expected from the heatmap showing all the 33 HPAP IMC-measured proteins in Revision 1, Sankey plots demonstrated that the removal of Myeloid and NK from

AnnoSpat's initialization had a marginal impact on predicted cell labels with specific antibodies and resulted in relabeling of almost all myeloid and NK cells as 'Unknown' (Figure S5C). Similarly, the inclusion of Myeloid from the Marker Protein file relabeled some of the 'Unknown' cells as 'Myeloid,' which exhibited high levels of CD68 expression as expected from a correct cell-type prediction (Figure S5D).

We stress that both sides of these Sankey plots present AnnoSpat-predicted and not true cell labels, as the latter information is not available and, for all practical purposes, cannot be obtained for 1.7 million cells included in our analysis. In the absence of true cell labels, Sankey plots cannot discern the correctness of predicted labels unless we examine all 33 HPAP IMC-measured proteins for each cell to determine whether the protein expression profiles of algorithmically-predicted cell labels match biologically-defined expected expression profiles.

2) Question 2: Like previous Question, it is not sufficient to visualize the results based on marker protein expression only. Tools like Sankey diagram would better reveal how these parameters affect the cell annotation results.

Per Reviewer 1's request, we included Sankey plots for the initialization parameter q_{high} . As expected from our earlier heatmap analysis, Sankey plots showed that while a more stringent q_{high} (i.e. higher values) had marginal impact on performance, especially for abundant cell types, a more relaxed q_{high} (i.e. lower values) adversely impacted "cell-type aware" initialization and decreased the accuracy of cell-type prediction for both abundant and rare cell types (Figure S6H-J).

Similar to the situation in Question 1, both sides of these Sankey diagrams present predicted and not true cell labels. Hence, these diagrams (or any other analysis) cannot discern the correctness of predicted labels unless we examine all the 33 HPAP IMC-measured proteins to determine whether the protein expression profiles of algorithmically-predicted cell labels match biologically-defined expected expression profiles.

3) Question 4: 1) The KL results were not shown in Fig. 3 in the revised manuscript? 2) The comparison between expert and algorithm annotation is very interesting but the results are in fact a bit inconsistent. On one hand, the authors compared the performance of different annotation results based on their KL divergence to the expert annotation while on the other hand they questioned the expert annotation results when the KL divergence showed discrepancies between AnnoSpat and expert annotations? The only conclusion that can be drawn from the current result should be "although AnnoSpat show better consistency with expert annotation, close examination shows that mislabeling can happen in expert annotations"

We thank the reviewer for pointing out this oversight. We now included supplementary Table S13 listing KL divergent results.

Following Reviewer 1 suggestion, we also revised the text.

Reviewer #3

The Authors in their revised version of the paper have addressed all the previous comments made by myself as well as by the other reviewers and they decided to perform new experiments as well as to amend the whole manuscript considerably improving it.

We are thrilled to learn that the Reviewer 2 concluded that we have addressed all the previous comments made by him/her as well as by the other reviewers and recommended our manuscript for publication.

Now data are significantly more robust and the paper is stronger and deeper than the previous version.

The paper is very accurate either in the performed analyses than in the new added

paragraphs, taking into account that they have included cell-type annotation in two new ICM experiments from the most recent Human pancreas Analysis Program donors.

It is also very important that they have included comparative analysis with HistoCAT. All the new Figures and the whole revised manuscript provide higher significance to the study.

Specifically, regarding the role of microvessels and their alterations during T1D, the authors have revised the manuscript to discuss this issue and the fact that blood flow impacts islet cells and their survival. These issues will be solved in a near future being now a limitation of the technique.

This referee acknowledges all the amendments and this new version of the manuscript.

Only some letters were lost in the written manuscript (spelling).

Reviewers' Comments:

Reviewer #1:

Remarks to the Author:

This manuscript has been substantially improved with the additional experiments and results. In particular, the reviewer agrees that lacking of ground truth in single-cell experiments such as IMC or CODEX makes it extremely challenging to develop and evaluate cell-type annotation methods. Nevertheless, with the additional evidence it now demonstrates that the proposed method indeed improves existing methods in terms of speed, number of annotated cells, and consistency, and annotation results match better with the cell biology. Based on that, the reviewer now recommends publishing the manuscript.